# HotProtein: A Novel Framework for Protein Thermostability Prediction and Editing

**Tianlong Chen**[*], **Chengyue Gong**[*]**, Daniel Jesus Diaz, Xuxi Chen, Jordan Tyler Wells,
Qiang Liu, Zhangyang Wang, Andrew Ellington, Alex Dimakis, Adam Klivans**
The University of Texas at Austin
{tianlong.chen,cygong17,danny.diaz,xxchen,jordantwells}@utexas.edu
{lqiang,atlaswang,andy.ellington,dimakis,klivans}@utexas.edu

## Abstract

The molecular basis of protein thermal stability is only partially understood and has major significance for drug and vaccine discovery. The lack of datasets and standardized benchmarks considerably limits learning-based discovery methods. We present `HotProtein`, a large-scale protein dataset with *growth temperature* annotations of thermostability, containing 182K amino acid sequences and 3K folded structures from 230 different species with a wide temperature range $-20°C \sim 120°C$. Due to functional domain differences and data scarcity within each species, existing methods fail to generalize well on our dataset. We address this problem through a novel learning framework, consisting of (1) Protein structure-aware pre-training (SAP) which leverages 3D information to enhance sequence-based pre-training; (2) Factorized sparse tuning (FST) that utilizes low-rank and sparse priors as an implicit regularization, together with feature augmentations. Extensive empirical studies demonstrate that our framework improves thermostability prediction compared to other deep learning models. Finally, we introduce a novel editing algorithm to efficiently generate positive amino acid mutations that improve thermostability. Codes are available in https://github.com/VITA-Group/HotProtein.

## 1 Introduction

Proteins are the bio-polymers responsible for executing most biological phenomena and, through evolution, have had their sequences optimized to carry out specific functions within specific cellular environments. A protein's stability is a multi-dimensional property that depends on a series of factors (Pucci et al., 2017; Cao et al., 2019) such as pH, salinity, and temperature (thermostability shown in Figure 1), making it hard to adapt a pro-

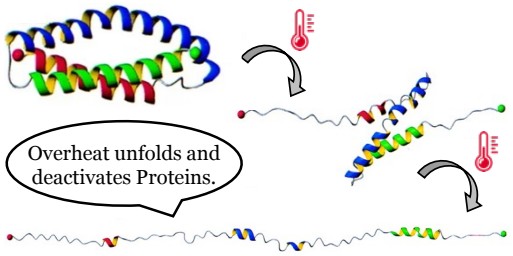

Figure 1: Overheat unfolds and deactivates proteins (Paci & Karplus, 2000).

tein to function outside of its endogenous cellular environment. Protein engineering is the field where natural proteins are mutated to improve their stability in exogenous environments and their overall fitness for a particular function. In protein engineering, one of the initial goals for most engineering campaigns is to improve the thermal stability of protein (Haki & Rakshit, 2003; Bruins et al., 2001; Frokjaer & Otzen, 2005). Thermally stabilized proteins are more robust and therefore enable downstream applications in the food (Kapoor et al., 2017), biofuel (Huang et al., 2020), detergent (Von der Osten et al., 1993), chemical (Cho et al., 2015), and pharmaceutical industry (Amara, 2013), drug design (De Carvalho, 2011; Mora & Telford, 2010), and bioremediation of environmental pollutants (Lu et al., 2022; Alcalde et al., 2006). Thus, to accelerate the engineering of a target protein it is critical to understand and accurately predict thermal stability changes of mutations. There has been a substantial effort from the community to quantitatively understand and model protein thermostability (*e.g.,* Pucci et al., 2017; Cao et al., 2019; Pucci & Rooman, 2014; Li et al., 2019; Pucci & Rooman, 2017; Pouyan et al., 2022). However, the generalizability of them is still unsatisfactory, and laborious experimental methods such as directed evolution are often preferred.

---

[*]Equal Contribution.

To enhance the capabilities of learning-based approaches, we present a large-scale, standardized protein benchmark, *i.e.,* `HotProtein`, with *organism-level* temperature annotations which is a lower bound of protein's melting temperature (Jarzab et al., 2020). It consists of 182K protein sequences and 3K folded structures from 230 different species, covering a broad temperature range of $-20°$C $\sim 120°$C. However, similar to Cao et al. (2019), naively trained deep models even on our dataset do not enable generalization to unseen proteins. The presumed reasons are (1) the considerable functional heterogeneity in proteins that arise from the environmental conditions and evolutionary history and (2) the scarcity of high-quality thermostability experimental data due to the massive cost and labor required to generate such data.

To tackle these pain points, we introduce a novel algorithmic pipeline to improve thermostability prediction. First, we enrich our sequence embeddings by infusing 3D structural information in a contrastive manner–we call this structure-aware pre-training (SAP). Then, we further fine-tune our model with a factorized sparse tuning (FST) approach. Here, we utilize factorized low-rank and sparse priors as implicit regularizers and leverage feature augmentation, such as mix-up (Verma et al., 2019) and worse-case augmentations (Chen et al., 2021d). FST greatly boosts the performance of tuned predictors, suggesting improved data efficiency and robustness against domain shifts (Li et al., 2022b; Chen et al., 2021c). Extensive evaluations on both `HotProtein` and the other existing protein datasets (*i.e.*, FireProtDB (Stourac et al., 2021)) verify our proposals' effectiveness.

Finally, to identify the top mutational predictions likely to improve thermal stability for a target protein, we develop a new optimization-based editing framework on top of a classifier or regressor, that attempts to mimic the process of directed evolution while limiting the stochasticity (Pucci & Rooman, 2014). Unlike existing protein engineering approaches (Eijsink et al., 2005; Couñago et al., 2006; Wijma et al., 2013) that directly utilize the predictions to generate mutational designs, our proposal maximizes the model's objective to approach a more thermostable label to identify input mutated sequences. Our contributions can be summarized as follows:

⋆ We collect and present a large-scale protein dataset, *i.e.,* `HotProtein`, with organism-level temperature annotations. We use the organism's environmental growth temperature to label and classify all proteins within each organism, which we use for thermostability prediction and editing. It contains 182K amino acid sequences and 3K folded 3D structures of proteins from 230 different species, covering five thermostability types, *e.g.,*, Cryophilic, Psychrophilic, Mesophilic, Thermophilic, and Hyperthermophilic.

⋆ We introduce a protein structure-aware pre-training by injecting 3D structural information into sequence embeddings in a contrastive fashion. It enhances the diversity and expressivity of the protein representations, resulting in improved thermostability predicting performance.

⋆ We introduce a robust and data-efficient tuning framework that performs weight updates in the factorized and sparse subspace together with augmented feature embedding. This leads to substantial performance improvements against data scarcity and severe distribution shifts.

⋆ We formulate the search for thermal stabilizing mutations as an optimization problem: for a target protein and a trained predictor, we customize an editing framework that optimizes the input protein sequences to identify thermostabilizing mutations.

⋆ Extensive experiments conducted on both thermostability prediction and protein editing tasks, consistently demonstrate the superiority of our proposals over various existing approaches (Rives et al., 2021). For example, when fine-tuned on experimentally determined $T_m$ dataset, Fire-ProtDB, our editing suggester achieves $53.93\%$ ($\uparrow 8.96\%$) precision in positive mutation classification, $50.79$ ($\uparrow 6.54$) Spearman $\rho$ correlation coefficient in the temperature regression, and $54.24\%$ ($\uparrow 1.83\%$) successful rate in generating positive single mutations.

## 2 RELATED WORKS

**Protein Thermostability Prediction.** To enhance a protein's stability, $\Delta\Delta$G and $\Delta T_m$ are common metrics by molecular biologists, enzymologists, and protein engineers. $\Delta\Delta$G evaluates the changes in free energy between a protein and a mutated variant. While $\Delta T_m$ evaluates the change in thermal tolerance between two protein variants. The two are related through the Van 't Hoff equation (Wright et al., 2017) and it is common to obtain $\Delta\Delta$G from $T_m$ measurements (*e,g,* Chen et al., 2013; Capriotti et al., 2005; Rodrigues et al., 2018; Pires et al., 2014a; Parthiban et al., 2006).

Most studies lack accurate large-scale thermostability data. For example, deepDDG (Cao et al., 2019) is trained on $5,766$ manually-curated $\Delta\Delta G$ measurements across 242 proteins, while one of their test set contains 173 experimental melting temperature changes ($\Delta T_m$) to assess how well deepDDG correlated with $\Delta T_m$ (Cao et al., 2019). Previously, researchers use empirical physics-based energy contributions (Kellogg et al., 2011), torsion angles (Parthiban et al., 2006), or graph-based distance patterns (Pires et al., 2014b) as features and apply different models, *e.g.,* physical models, residue interaction networks (Giollo et al., 2014), SVMs (Chen et al., 2013), to predict the thermodynamics $\Delta\Delta G$. However, none of these methods have shown strong generalization.

**Protein Engineering.** Protein engineering with machine learning is usually formulated as an energy-guided refinement process by maximizing a pre-defined energy function with changing input data (*e.g.,* torsion angle, 1D amino acid sequence, or 2D contact map) (AlQuraishi, 2019; Kuhlman et al., 2003; Huang et al., 2016; Kuhlman & Bradley, 2019). A group of works trains a generative model or autoencoder and then optimizes the continuous hidden representation to maximize some given objectives (*e.g.,* Gligorijevic et al., 2021; Shuai et al., 2021; Hawkins-Hooker et al., 2021; Hoffman et al., 2022). Another category of works train models to map the input data to the target property (e.g. temperature, energy, etc.), and then optimize the discrete input space with combinatorial optimization (Norn et al., 2021). These methods have been applied to different kinds of input data, *e.g.,* 1D structural features (Norn et al., 2021; Wang et al., 2017; Karplus, 2009), torsion angles, and contact maps (Jones & Kandathil, 2018; del Alamo et al., 2021).

**Description about more related works.** Due to space limitations, we place discussions about protein engineering background, directed evolution, guided directed evolution, other protein thermostability datasets, sequence-/structure-based protein models, and sparse and low-rank subspace fine-tuning in our Appendix A.

## 3 THE HOTPROTEIN DATASET

To obtain thermostable labels for an organism's proteome, we collect the raw data from the NCBI BioProject (Barrett et al., 2012), which offers an organizational framework to access the (meta-)data about research projects, which is deposited or planned for deposition, into archival repositories.

**Preprocess.** From the NCBI bioproject XML file[1], we filter organisms where the environmental data (*i.e.,* "OptimumTemperature" and "TemperatureRange") is available. After removing duplicate organism entries (keeping the first entry), this provides us with $1,733$ unique entries. Next, we proceed to download these organisms' proteomes from UniProt[2] via their taxids and bin the proteomes based on the "TemperatureRange" classification of that organism. Finally, we remove all proteins over $1,500$ amino acids in length and utilize CD-Hit[3] to cluster protein sequences across organisms within a "TemperatureRange" class at a sequence similarity threshold of $50\%$. In each cluster, we only keep proteins whose sequence lengths are between $200 \sim 550$.

**Annotation and Folding.** To further increase the fidelity of our annotations, we remove organisms from each "TemperatureRange" bin where the "OptimumTemperature" does not fall within the corresponding limits: ❶ *Hyperthermophilic* ($> 75$ Celsius), ❷ *Thermophilic* ($45 \sim 75$ Celsius), ❸ *Mesophilic* ($25 \sim 45$ Celsius), ❹ *Psychrophilic* ($5 \sim 25$ Celsius), and ❺ *Cryophilic* ($-20 \sim 5$ Celsius). The filtered proteomes and their corresponding annotations form the HotProtein dataset and are utilized throughout the classification and regression tasks in our study.

For a random subset of the HotProtein dataset ($\sim$ 3K), we predict structure with AlphaFoldV2 (AlQuraishi, 2019; Jumper et al., 2021) to obtain their 3D coordinates. The official implementation is adopted. We report the predicted template modeling (P-TM) scores for each folded structure in Figure 2 A.3) and B.3). The P-TM score is a rough approximation of the folding quality (AlQuraishi, 2019), and we kept structures only with a P-TM $\geq 0.8$.

**Properties.** As described in Figure 2, we generate four distinct testbeds from the HotProtein dataset that differed in scale: (1) HP-S$^2$C2 has 1026 "hot" ($\geq 45°C$) and 939 "cold" ($< 45°C$) proteins from 61 and 4 species, respectively. Both sequence and structure statistics of these proteins are provided. (2) HP-S$^2$C5 consists of both sequences and structures for $\{73, 387, 195, 196, 189\}$

---

[1] `https://ftp.ncbi.nlm.nih.gov/bioproject/bioproject.xml`
[2] `https://www.uniprot.org/help/uniprotkb`
[3] `http://weizhong-lab.ucsd.edu/cd-hit/`

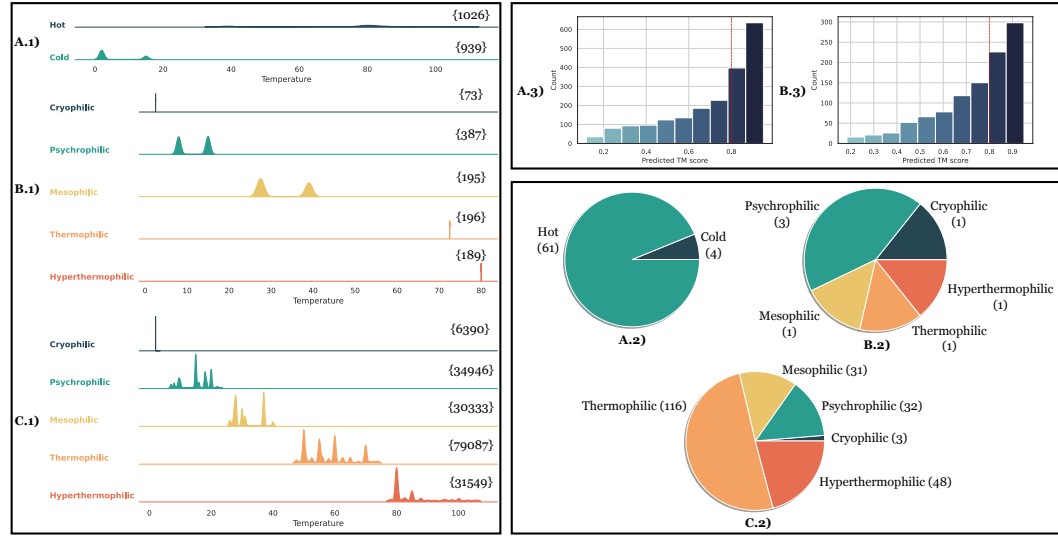

Figure 2: The overview of `HotProtein` dataset. Figures A.1 $\sim$ 3), B.1 $\sim$ 3), and C.1 $\sim$ 2) collect the statistics of `HP-S`$^2$`C2`, `HP-S`$^2$`C5`, and `HP-S` respectively. The *left* figure records the density distribution of each category over the protein's **growth (organism-level) temperature**. The density in a bin is computed as $\frac{\text{\# proteins within the bin}}{\text{\# proteins in the category}}$. {1026} indicates that there are 1026 proteins in the corresponding class "Hot". The *upper right* figure is the predicted template modeling (P-TM) score from AlphaFoldv2 (AlQuraishi, 2019; Jumper et al., 2021), reflecting the quality of folded protein structures. A larger P-TM suggests a better quality and P-TM $\geq 0.8$ (red lines) is a normal threshold for satisfied folded structures. The *bottom right* figure presents the species distribution of the three datasets, and # species in certain categories is included in the brackets.

proteins sampled from the five categories, from *Cryophilic* to *Hyperthermophilic*. (3) `HP-S` is the entire sequence `HotProtein` dataset. It contains {6390, 34946, 30333, 79087, 31549} sequences from {3, 32, 31, 116, 48} different species, of five classes ordered from *Cryophilic* to *Hyperthermophilic*. (4) Moreover, `HP-SC2` as a 2-class variant is created by merging *Hyperthermophilic* and *Thermophilic* as "hot" class and the other three as "cold" class. **All temperature annotations are organism-level, serving as a lower bound of for a protein's melting temperature**. Given their number of samples, `HP-S`$^2$`C2/C5` and `HP-S/SC5` are regarded as small- and large-scale datasets.

## 4 METHODOLOGY

Denote the thermostability dataset by $\mathcal{D} = \{\boldsymbol{x}_i, y_i\}_{i=1}^N$, where $\boldsymbol{x}_i$ stands for the input while $y_i$ is the thermostability critical temperature (either real value or class index). Here we describe 1) how we pretrain the model $\mathcal{F}$, and 2) how we finetune to pretrained model to fit $\mathcal{D}$.

### 4.1 PROTEIN STRUCTURE-AWARE PRE-TRAINING

Previous protein pre-trained models (*e.g.,* Rives et al., 2019; Vig et al., 2020; Rao et al., 2021; Meier et al., 2021) mainly focus on masking prediction tasks and amino acid representations. These pre-trained models achieve considerable improvementscompared to traditional computation methods (*e.g.,* Schymkowitz et al., 2005; Montanucci et al., 2019; Chen et al., 2020) on amino acid prediction tasks, *e.g.,* contact prediction, mask prediction (Brandes et al., 2022), mutational effect prediction (Meier et al., 2021; Notin et al., 2022; Li et al., 2022a). On the other hand, directly learning universal protein embedding is potentially useful for protein prediction tasks. Inspired by recent sentence representation learning works (Gao et al., 2021) and 3D structure-aware pre-training (Hsu et al., 2022), we adopt a contrastive loss for representation learning.

**Contrastive Loss Design.** Denote the model as $\mathcal{F}$, our contrastive loss function is defined as:

$$\mathcal{L}_{\text{InfoNCE}}(\boldsymbol{x}_i, \text{Neg}(\boldsymbol{x}_i)) = -\log \frac{\exp(\mathcal{F}'(\boldsymbol{x}_i) \cdot \mathcal{F}''(\boldsymbol{x}_i)/\tau)}{\exp(\mathcal{F}'(\boldsymbol{x}_i) \cdot \mathcal{F}''(\boldsymbol{x}_i)/\tau) + Z(\boldsymbol{x}_i, \text{Neg}(\boldsymbol{x}_i))}, \qquad (1)$$

where $Z(\boldsymbol{x}_i, \text{Neg}(\boldsymbol{x}_i)) = \sum_{\boldsymbol{x}_j \in \text{Neg}(\boldsymbol{x}_i)} \exp(\mathcal{F}'(\boldsymbol{x}_i) \cdot \mathcal{F}'^{\text{mom}}(\boldsymbol{x}_j)/\tau)$. Here, $\mathcal{F}'(\boldsymbol{x}_i)$ and $\mathcal{F}''(\boldsymbol{x}_i)$ are two copies of randomly perturbed models by injecting independent dropout noises into $\mathcal{F}(\boldsymbol{x}_i)$, and $\text{Neg}(\cdot)$ presents a set of negative examples, and $\mathcal{F}^{\text{mom}}$ is a slowly updated momentum model

Figure 3: The overall pipeline of our proposals. The *left* describes the protein structure-aware pre-training (SAP); The *right* presents the factorized sparse tuning (FST).

(see below) and $\tau$ is the temperature value. Notice that random mutating amino acid could introduce unknown changes (Shortle, 2009; Resch et al., 2008), we add perturbations to hidden representations and inject layer-wise dropout noise (Srivastava et al., 2014) with a rate $0.05$ in practice, and follow the hyperparameter configurations in He et al. (2020).

**Leverage 3D Information.** We combine an additional 3D structure model using 3D inputs with the sequence-based model, to enhance the performance. We therefore slightly change Equation 1 into,

$$\mathcal{L}_{3D}(\boldsymbol{x}_i, \text{Neg}(\boldsymbol{x}_i)) = -\log \frac{\exp(\mathcal{F}'(\boldsymbol{x}_i) \cdot \mathcal{F}''_{3D}(\boldsymbol{x}_i)/\tau)}{\exp(\mathcal{F}'(\boldsymbol{x}_i) \cdot \mathcal{F}''_{3D}(\boldsymbol{x}_i)/\tau) + Z(\boldsymbol{x}_i, \text{Neg}(\boldsymbol{x}_i))}, \tag{2}$$

$$Z(\boldsymbol{x}_i, \text{Neg}(\boldsymbol{x}_i)) = \sum_{\boldsymbol{x}_j \in \text{Neg}(\boldsymbol{x}_i)} \left\{ \exp(\mathcal{F}'(\boldsymbol{x}_i) \cdot \mathcal{F}'^{\text{mom}}_{3D}(\boldsymbol{x}_j)/\tau) + \exp(\mathcal{F}'_{3D}(\boldsymbol{x}_i) \cdot \mathcal{F}'^{\text{mom}}(\boldsymbol{x}_j)/\tau) \right\},$$

where $\mathcal{F}_{3D}$ stands for the 3D structure model which uses 3D coordinates as inputs, $\mathcal{F}^{\text{mom}}_{3D}$ is its momentum model. The memory bank logs the negative examples from both the sequence and 3D models. In short, a pair for sequence and 3D model representation is regarded as the positive pair, and we aim at injecting 3D representation information into the sequence model.

## 4.2 FACTORIZED SPARSE TUNING UNDER DATA SCARCITY AND DISTRIBUTION SHIFT

In real-world application domains, it is extremely challenging or even infeasible to collect a sufficient large-scale (engineered) protein dataset, due to the massive time and resource cost of transformation, protein expression, and purification. Therefore, data scarcity is one of the crucial bottlenecks in predicting protein properties. Another obstacle lies in the substantial data distribution shifts. Proteins with different (sometimes even the same (Xia, 2021)) functionalities have distinctive and idiographic structures, sharing few common characteristics. To tackle these two issues, we introduce a factorized sparse tuning pipeline (FST). It leverages the low-rank and sparse priors as implicit regularizations for enhanced data-efficiency (*e.g.,* Khan & Stavness, 2019; Shalev-Shwartz & Ben-David, 2014; Rasmussen & Ghahramani, 2000; Zhou et al., 2018; Arora et al., 2018; Zhang et al., 2021) and robustness to domain shifts (Li et al., 2022b).

**Enforcing the Low-rank Prior through Factorization.** Given a pre-trained model $W_p \in \mathbb{R}^{m \times n}$, we perform a low-rank decomposition to its weight update $\Delta W$ represented as $W_p + \Delta W = W_p + UV$, where $U \in \mathbb{R}^{m \times r}$, $V \in \mathbb{R}^{r \times n}$, and the rank $r \ll \min\{d, k\}$. Usually, $r = 4$ or $8$ (*i.e.,* $0.6\%$ of total parameters) is sufficient to achieve a great performance in our case. During forward, the input is fed into both dense pre-training $W_p$ and its low-rank representation $UV$. The obtained features are summed in a coordinate-wise manner, as demonstrated in Figure 3 (*right*). Take the original feature $h = W_p \boldsymbol{x}$ as an example. Our modified features can be described as below:

$$\hat{h} = W_p \boldsymbol{x} + \Delta W \boldsymbol{x} = W_p \boldsymbol{x} + UV \boldsymbol{x}. \tag{3}$$

As for the backpropagation, $W_p$ is frozen and low-rank matrices $\{U, V\}$ receive gradient updates. We adopt Xavier normal (Glorot & Bengio, 2010) and zero initialization for $U$ and $V$ respectively, and therefore the low-rank update is zero in the beginning. As suggested by Hu et al. (2021), $UV \boldsymbol{x}$ can be scaled via an extra hyperparameter $\alpha$, which works similarly to the learning rate. In our case, we find $\alpha$ is not sensitive and set it as $4$, *i.e.,* the default value used in Hu et al. (2021).

**Enforcing the Sparsity Prior.** We introduce the sparsity into tuning processes as an implicit structural prior by modeling a sparse weight update with $S \in \mathbb{R}^{m \times n}$. As indicated in Figure 3, our refined features $\tilde{h}$ are depicted as follows:

$$\tilde{h} = W_p \boldsymbol{x} + (UV + S)\boldsymbol{x}, \quad S = \begin{cases} s_{i,j}, & (i,j) \in \Omega \\ 0, & (i,j) \in \Omega^{\text{C}} \end{cases}, \tag{4}$$

where a "residual" feature $S\boldsymbol{x}$ is point-wisely added to $\hat{h}$ from Equation 3, $i \in \{1, 2, \cdots, m\}$, and $j \in \{1, 2, \cdots, n\}$. The set $\Omega$ determines the position of trainable $s_{i,j}$ and pruned elements, where the latter is $0$ across the whole training. We compute the initial sparse matrix of $S$ by $i)$ first solving a robust principal component decomposition (Candès et al., 2011) of $W_p$ with an efficient algorithm (*i.e.,* GreBsmo (Zhou & Tao, 2013)), and $ii)$ then eliminating the elements with the least magnitude of obtained sparse solutions. In this way, step $i)$ produces the initial values of $s_{i,j}$ and step $ii)$ constructs the set $\Omega$, where we observe $|\Omega| = 64$ (*i.e.,* $0.01\%$ of total parameters) is good enough in our case. As also revealed in Yu et al. (2017), combining the low-rank and sparse weight updates is capable of delivering superior performance, compared to either of them.

**Feature Augmentation.** Another group of common fixes to data scarcity and domain shift problems is data augmentation (Shorten & Khoshgoftaar, 2019). However, most of existing data-level augmentations are unrealistic to protein sequences, including regional dropout (Zhou & Tao, 2013; Zhong et al., 2020) and CutMix (Yun et al., 2019) in computer vision; synonym replacement (Kolomiyets et al., 2011; Zhang et al., 2015; Wang & Yang, 2015), random insertion, swap, and deletion (Wei & Zou, 2019) in natural language processing. The reason is that even a single amino acid mutation for the protein sequence may dramatically change its functionality (Resch et al., 2008; Shortle, 2009). To avoid ambiguity, we utilize feature-level augmentations which manipulate the model's intermediate feature embedding. Specifically, we examine two effective mechanisms: (1) *Mixup* feature augmentation. It creates a fused feature $\tilde{h}_{\text{aug}} = \lambda \times \tilde{h}_1 + (1 - \lambda) \times \tilde{h}_2$ and its associated soft label $y_{\text{aug}} = \lambda \times y_1 + (1 - \lambda) \times y_2$, where $\{\tilde{h}_1, y_1\}$, $\{\tilde{h}_2, y_2\}$ are {feature embedding, label} of two different sequences and $\lambda = 0.2$ in our experiments. (2) *Worst-case* feature augmentation. It injects worst-case noises $\delta$ and builds an augmented feature $\tilde{h}_{\text{aug}} = \tilde{h} + \delta$, where $\delta$ is generated by $\max_\delta \mathcal{L}(\mathcal{F}(\boldsymbol{x}, \tilde{h} + \delta), y)$ with the gradient ascent (Chen et al., 2021d). $\mathcal{L}$, $\mathcal{F}$, and $\boldsymbol{x}$ denote the objective function, model, and input sequence, respectively.

# 5 EXPERIMENTS

## 5.1 IMPLEMENTATION DETAILS

**Metrics.** The performance is evaluated on the test splits. {Accuracy, Precision} and {Spearman, Pearson} correlation coefficients are used for classification and regression tasks. For `HP-S`$^2$`C2` and `HP-S`$^2$`C5`, 10-fold evaluation is conducted; while on `HP-S` and `HP-SC2`, we run three replicates with different random seeds. Average performance and its $95\%$ confidence intervals are reported.

**Training Details.** *Baselines.* 3D GCN (Gligorijević et al., 2021) is trained for 20 epochs, with an initial learning rate of $1 \times 10^{-4}$ that decays by $0.1$ at the 10th epoch. For TAPE (Rao et al., 2019), we train it for $4$ epochs, with an initial learning rate of $1 \times 10^{-4}$ and a linear decay schedule. As for ESM-1B, we follow (Rives et al., 2021) and only train a linear classification head on the top of ESM-1B backbone. The head tuning consists of $4$ epochs with an initial learning rate of $2 \times 10^{-2}$ and an OneCycle (Smith & Topin, 2019) decay scheduler. A training batch size of $4$ is used across all experiments. Since we start tuning from pre-trained models (Rao et al., 2019; Rives et al., 2021), the performance of TAPE and ESM-1B are usually saturated after $2 \sim 3$ epochs.

▷ *SAP.* We use AlphaFoldDB (Jumper et al., 2021) for SAP protein pre-training. We filter the data with sequence length and data quality and finally get 270K data. ESM-1B (Rives et al., 2019) and ESM-IF (Hsu et al., 2022) backbone are used to process the sequence and 3D coordinate inputs, and an average pooling layer is applied to the final-layer token representations of ESM models and get protein embeddings. A momentum encoder with $\tau = 1.0$, momentum encoder coefficient $\alpha = 0.9999$ and memory bank of size $65,536$ is used and the model is trained for $4$ epochs, with AdamW optimizer, weight decay $10^{-12}$, batch size $512$ and an initial learning rate $10^{-6}$ decayed with OneCycle (Smith & Topin, 2019) decay scheduler.

▷ *FST.* For our FST, we choose an initial learning rate of $1 \times 10^{-2}$ for the linear classification head, and an initial learning rate of $1 \times 10^{-3}$ for training the low-rank and sparse components in ESM-1B. Other training configurations inherit the same ones from tuning ESM-1B. As for the hyperparameters of rank $r$ and the number of non-zero elements $|\Omega|$ in FST, we perform screenings on $r \in \{4, 8, 16\}$ and $|\Omega| \in \{16, 32, 64, 128\}$, where we choose $(r, |\Omega|) = (4, 64)$ on `HP-S`$^2$`C2/C5` and $(r, |\Omega|) = (8, 64)$ on `HP-S` and `HP-SC2`. Meantime, we adopt a one-step gradient ascent with a step size of $1 \times 10^{-5}$ to generate worst-case feature augmentations, and apply them to the last two layers of ESM-1B, as suggested in Chen et al. (2021d).

Table 1: Performance of predicting thermostability with classification. Accuracy (%) is reported for all three datasets, and Precision (%) is calculated for the 2-class classification on HP-S$^2$C2 and HP-SC2. "FST", "Aug.", and "SAP" denote factorized sparse tuning, feature augmentation, and protein structure-aware pre-training, respectively. FST adopts $(r, |\Omega|) = (4, 64)$ on HP-S$^2$C2/C5 and $(r, |\Omega|) = (8, 64)$ on HP-S/SC2. N.A. means "not applicable". 95% confidence interval are computed via the 10-fold evaluation on HP-S$^2$C2/C5 and 3 replicates on HP-S/SC2.

| Methods | HP-S²C2 | | HP-S²C5 | HP-S | HP-SC2 | |
|---|---|---|---|---|---|---|
| | Accuracy | Precision | Accuracy | Accuracy | Accuracy | Precision |
| 3D GCN (Gligorijević et al., 2021) | 78.88±1.57 | 73.39±2.76 | 67.40±2.11 | N.A. | N.A. | N.A. |
| TAPE (Rao et al., 2019) | 83.31±1.10 | 76.42±3.06 | 66.44±2.30 | 64.75±0.23 | 76.37±0.25 | 80.64±0.50 |
| ESM-IF1 (Hsu et al., 2022) | 79.08±0.85 | 76.49±3.96 | 58.75±2.46 | N.A. | N.A. | N.A. |
| ESM-1B (Rives et al., 2021) | 91.19±0.47 | 84.18±1.71 | 83.26±1.54 | 69.50±0.16 | 86.24±0.22 | 88.14±1.62 |
| ESM-1B + Aug. | 91.74±0.79 | 86.09±2.12 | 84.32±1.41 | 69.54±0.39 | 86.26±0.22 | 88.27±1.43 |
| ESM-1B + FST | 91.85±0.45 | 84.85±1.04 | 85.96±1.13 | 72.97±0.28 | 87.50±0.12 | 88.71±1.73 |
| ESM-1B + FST + Aug. | 91.91±0.64 | 86.10±1.14 | 86.08±1.33 | 73.09±0.10 | 87.54±0.38 | 88.83±1.27 |
| ESM-1B + FST + Aug. + SAP | **92.36±0.58** | **86.51±1.67** | **86.25±1.03** | **73.21±0.13** | **87.57±0.10** | **89.07±1.29** |

Table 2: Performance of thermostability regression. Correlation coefficients are reported for all three datasets. 95% confidence interval are computed via the 10-fold evaluation on HP-S$^2$C2/C5 and 3 replicates on HP-S.

| Methods | HP-S²C2 | | HP-S²C5 | | HP-S | |
|---|---|---|---|---|---|---|
| | Spearman | Pearson | Spearman | Pearson | Spearman | Pearson |
| 3D GCN (Gligorijević et al., 2021) | 0.490±0.019 | 0.469±0.019 | 0.291±0.053 | 0.301±0.074 | N.A. | N.A. |
| TAPE (Rao et al., 2019) | 0.432±0.061 | 0.386±0.065 | 0.367±0.063 | 0.364±0.047 | 0.504±0.013 | 0.453±0.031 |
| ESM-IF1 (Hsu et al., 2022) | 0.589±0.040 | 0.547±0.036 | 0.373±0.036 | 0.377±0.035 | N.A. | N.A. |
| ESM-1B (Rives et al., 2021) | 0.890±0.018 | 0.893±0.0238 | 0.712±0.043 | 0.804±0.023 | 0.807±0.001 | 0.809±0.001 |
| ESM-1B + Aug. | 0.895±0.014 | 0.909±0.010 | 0.714±0.034 | 0.811±0.034 | 0.808±0.002 | 0.809±0.001 |
| ESM-1B + FST | 0.898±0.008 | 0.900±0.009 | 0.742±0.039 | 0.815±0.024 | 0.819±0.002 | 0.825±0.004 |
| ESM-1B + FST + Aug. | 0.892±0.011 | 0.912±0.013 | 0.747±0.026 | 0.818±0.025 | 0.820±0.001 | 0.825±0.003 |
| ESM-1B + FST + Aug. + SAP | **0.906±0.010** | **0.923±0.012** | **0.754±0.035** | **0.837±0.019** | **0.823±0.001** | **0.827±0.003** |

## 5.2 PREDICTING THERMOSTABILITY VIA CLASSIFICATION AND REGRESSION

**Comparison to Existing Approaches.** We evaluate our proposals and compare with existing approaches on HP-S$^2$C2/C5 and HP-S/SC2 for both classification and regression tasks. 3D GCN (Gligorijević et al., 2021) and TAPE (Rao et al., 2019) are classical structure- and sequence-based models, while ESM-IF1 (Hsu et al., 2022) and ESM-1B (Rives et al., 2021) emerges recently as current state-of-the-art approaches, dealing with structure and sequence inputs respectively. From results shown in Table 1 and 2, several consistent observations can be drawn: ❶ *Compared with baselines.* Our proposal, *i.e.,* ESM-1B+FST+Aug.+SAP, greatly surpasses various baseline by a margin of {1.17% ~ 13.48% accuracy, 2.33% ~ 13.12% precision, 0.016 ~ 0.474 Spearman, and 0.030 ~ 0.537 Pearson correlation} on HP-S$^2$C2, {2.99% ~ 27.50% accuracy, 0.042 ~ 0.463 Spearman, and 0.033 ~ 0.536 Pearson correlation} on HP-S$^2$C5, {3.71% ~ 8.46% accuracy, 0.016 ~ 0.319 Spearman, and 0.018 ~ 0.374 Pearson correlation} on HP-S, and {1.33% ~ 11.2% accuracy and 0.93% ~ 8.43% precision on HP-SC2. These generalization improvements on hold-out proteins validate the effectiveness of our methods in tackling the functional domain shift and data-scarcity issues, *i.e.,* pre-training on a general-purpose dataset UniRef50 (22M samples) and tuning on the specific-domain dataset HotProtein (182K samples). Moreover, our methods perform much stably in general, evidenced by the reduced confidence interval of multiple runs. ❷ *Structure versus sequence-based models.* On the classification task, sequence-based models like ESM-1B and TAPE show clear performance advantages in most cases. As for the regression task, ESM-1B achieves an overwhelming superiority among the four baseline approaches, while the next best model is ESM-IF1 which takes protein structures as inputs. It suggests that powerful mechanisms for utilizing 3D structure information are still missing on our challenging HotProtein dataset. We make a pioneer attempt by leveraging 3D information to enhance sequence-based models. ❸ *Does SAP, FST, and Aug helps?* We examine these three components in an incremental manner and we observe: $i$) the performance gains from SAP demonstrate the benefits of treating 3D protein structures as auxiliary information; $ii$) Both Aug. & FST strengthen the model tuning and a combination of them enjoys extra improvements; $iii$) Among these ingredients of our proposal, FST contributes the most to superior performance. Specifically, Aug consistently obtains improvements in terms of the average performance; FST usually leads to statistically significant improvements with respect to the 95% confidence interval, especially on the HP-S/SC2 dataset. ❹ *Small v.s. large datasets.* Feature level augmentations are more beneficial at small-scale datasets, while FST and SAP bring performance gains for both small (HP-S$^2$C5) and large (HP-S) datasets.

Table 3: Ablation study on the components of our framework. Accuracy (%) for classification and Spearman & Pearson correlation coefficients for regression are reported. 95% confidence interval are computed via the 10-fold evaluation on `HP-S²C5` and 3 replicates on `HP-S`.

| Methods | | HP-S²C5 | | | HP-S | | |
|---|---|---|---|---|---|---|---|
| | | Accuracy | Spearman | Pearson | Accuracy | Spearman | Pearson |
| Based on ESM-1B (Rives et al., 2021) | | $83.26_{\pm1.54}$ | $0.712_{\pm0.043}$ | $0.804_{\pm0.022}$ | $69.50_{\pm0.16}$ | $0.807_{\pm0.001}$ | $0.809_{\pm0.001}$ |
| Feature Aug. | Random | $82.97_{\pm1.46}$ | $0.708_{\pm0.043}$ | $\mathbf{0.814_{\pm0.023}}$ | $69.46_{\pm0.28}$ | $0.807_{\pm0.001}$ | $\mathbf{0.809_{\pm0.001}}$ |
| | Mixup | $83.94_{\pm1.43}$ | $\mathbf{0.720_{\pm0.042}}$ | $0.810_{\pm0.027}$ | $69.27_{\pm0.27}$ | $0.805_{\pm0.001}$ | $0.806_{\pm0.003}$ |
| | Worst-case | $\mathbf{84.32_{\pm1.41}}$ | $0.714_{\pm0.034}$ | $0.811_{\pm0.034}$ | $\mathbf{69.54_{\pm0.39}}$ | $\mathbf{0.808_{\pm0.002}}$ | $0.809_{\pm0.001}$ |
| # Rank in FST | $r=4$ | $\mathbf{85.96_{\pm1.13}}$ | $\mathbf{0.742_{\pm0.039}}$ | $\mathbf{0.815_{\pm0.024}}$ | $72.75_{\pm0.17}$ | $0.818_{\pm0.001}$ | $0.821_{\pm0.002}$ |
| | $r=8$ | $85.00_{\pm2.16}$ | $0.703_{\pm0.032}$ | $0.801_{\pm0.017}$ | $72.97_{\pm0.28}$ | $0.819_{\pm0.002}$ | $\mathbf{0.825_{\pm0.004}}$ |
| | $r=16$ | $83.75_{\pm1.92}$ | $0.725_{\pm0.040}$ | $0.788_{\pm0.019}$ | $\mathbf{73.20_{\pm0.40}}$ | $\mathbf{0.821_{\pm0.001}}$ | $0.825_{\pm0.002}$ |
| Sparsity in FST | $|\Omega|=16$ | $85.19_{\pm1.02}$ | $0.702_{\pm0.034}$ | $0.794_{\pm0.027}$ | $72.82_{\pm0.43}$ | $0.815_{\pm0.004}$ | $0.822_{\pm0.005}$ |
| | $|\Omega|=32$ | $85.57_{\pm1.16}$ | $0.706_{\pm0.035}$ | $0.767_{\pm0.060}$ | $72.85_{\pm0.49}$ | $0.818_{\pm0.002}$ | $0.820_{\pm0.002}$ |
| | $|\Omega|=64$ | $\mathbf{85.96_{\pm1.13}}$ | $\mathbf{0.742_{\pm0.039}}$ | $\mathbf{0.815_{\pm0.024}}$ | $\mathbf{72.97_{\pm0.28}}$ | $\mathbf{0.819_{\pm0.002}}$ | $\mathbf{0.825_{\pm0.004}}$ |
| | $|\Omega|=128$ | $85.79_{\pm1.08}$ | $0.729_{\pm0.042}$ | $0.807_{\pm0.029}$ | $72.75_{\pm0.49}$ | $0.818_{\pm0.004}$ | $0.822_{\pm0.006}$ |
| 3D infor. in SAP | w.o. | $83.89_{\pm1.10}$ | $0.718_{\pm0.023}$ | $0.815_{\pm0.019}$ | $69.80_{\pm0.48}$ | $0.808_{\pm0.001}$ | $0.811_{\pm0.002}$ |
| | w. | $\mathbf{85.58_{\pm1.42}}$ | $\mathbf{0.727_{\pm0.035}}$ | $\mathbf{0.817_{\pm0.022}}$ | $\mathbf{71.52_{\pm2.29}}$ | $\mathbf{0.815_{\pm0.001}}$ | $\mathbf{0.819_{\pm0.003}}$ |

**Ablation Study.** A comprehensive ablation is presented in Table 3, where we inspect different feature augmentations, the rank number in FST, the sparsity in FST, and the necessity of 3D information in SAP, on top of the vanilla ESM-1B model. ❶ *Diverse feature augmentations.* Besides mixup and worst-case feature augmentations, a straightforward baseline ("Random") that applies a Gaussian noise from $\mathcal{N}(0, 0.1^2)$ to features, is also implemented. We see: random feature augmentation usually incurs a performance degradation; mixup benefits models tuned on the small-scale dataset (`HP-S²C5`), while hurts on the large-scale dataset (`HP-S`); worst-case feature augmentations consistently improve the generalization of tuned models on both `HP-S²C5` and `HP-S`, which is adopted by default in all other experiments of Table 1, 2, and 4. ❷ *The # rank in FST.* The larger dataset prefers FST with a higher rank such as $r=8$ or 16, compared to the smaller dataset in which FST with $r=4$ works the best. The finding coincided with the ones in Hu et al. (2021); Chen et al. (2021e). ❸ *The sparsity in FST.* FST with $|\Omega|=64$ is a "sweet point". Superfluous tuning elements in $\Omega$ may lead to inferior results. ❹ *3D infor. in SAP.* Directly introducing a contrastive loss in Equation 1 to the pre-training, has already boosted ESM-1B's performance, implying an improved protein embedding learning. Coupling the 3D information of protein structures, obtains an additional quality bonus for the pre-training.

**More ablation studies.** We summarize the results of additional ablation studies here and refer readers to the Appendix D for details: ❶ We achieve the best results on extra test benchmarks and class-balanced test sets. ❷ SAP outperforms other approaches to inject structure information. ❷ Training from scratch yields worse results than pretraining. ❸ On ESM-1B, we test several other baselines. We notice that additional $\ell_2$ or $\ell_1$ regularization, end-to-end fine-tuning, partially frozen tuning, and other feature aggregation methods all come to worse results than our current ESM-1B.

## 5.3 PROTEIN EDITING TOWARDS IMPROVED THERMOSTABILITY

Based on the models we trained in previous sections, three approaches for protein editing are proposed. We first describe the setting we use to edit proteins, and then report the performance of different models for three settings, classification, regression, and `Editing`. To demonstrate the effectiveness of protein editing towards improved thermostability, we further evaluate our proposals on FireProtDB[4] which is a manually curated database of the protein stability data for single-point mutants. It contains over 200 natural protein amino acid sequences, *i.e.,* $\mathcal{P} = \{p^{(i)}\}, i \in \{1, \cdots, x\}$, and their 3.9K mutated sequences, *i.e.,* $\hat{\mathcal{P}} = \{\hat{p}_j^{(i)}\}, i \in \{1, \cdots, x\}$ and $j \in \{1, \cdots, n^{(i)}\}$, where $n^{(i)}$ is the number of mutated variants for the original protein sequence $p^{(i)}$. We take $\mathcal{P}$ as the training set and $\hat{\mathcal{P}}$ as the hold-out testing set. Detailed configurations are referred to the Appendix C.

**Editing Suggestions via Classifiers.** We deliver the protein editing suggestion via proposed classifiers. The five-class classification is performed in both fine-tuning and testing stages with $\mathcal{P}$ and $\hat{\mathcal{P}}$, respectively. Furthermore, we regard a mutation as *positive* if $\hat{p}_j^{(i)}$ is predicted to a class that has higher temperatures than the category of its original counterpart $p^{(i)}$ (*e.g.,* from *Psychrophilic*

---

[4]`https://loschmidt.chemi.muni.cz/fireprotdb/`

Table 4: Evaluation of protein editing suggestions. Accuracy (%) and Precision (%) for classifiers, Kendall $\tau$ & Spearman $\rho$ rank correlation coefficients for regressors, and successful rate (%) for adversarially learned mutations of protein sequences are reported. 95% confidence interval is computed via three trials.

| FireProtDB | | Classifier | | Regressors | | Learned Mutations |
|---|---|---|---|---|---|---|
| | | Accuracy | Precision | Kendall $\tau$ | Spearman $\rho$ | Successful Rate |
| Zero shot | TAPE | 44.75 | 30.18 | 0.07 | 5.64 | 34.78 |
| | ESM-1B | 62.26 | 41.73 | 8.34 | 9.49 | 40.39 |
| | Ours | 66.18 | 43.60 | 13.65 | 19.96 | 43.46 |
| Fine-tune on $\mathcal{P}$ | TAPE | 55.18±0.38 | 38.85±0.36 | 32.33±1.35 | 43.78±1.87 | 44.59±0.45 |
| | ESM-1B | 69.30±0.53 | 44.97±0.47 | 34.76±1.42 | 44.25±1.89 | 52.41±0.38 |
| | Ours | **71.42**±**0.34** | **53.93**±**0.39** | **36.97**±**0.78** | **50.79**±**1.22** | **54.24**±**0.41** |

to *Mesophilic*); otherwise, we label the mutation as *negative*. Then, we compute the associated accuracy and precision to evaluate the quality of editing suggestions from our proposals.

**Editing Suggestions via Regressors.** We can also suggest possible editing via regressors. Specifically, we fine-tune models to regress the protein's temperature in $\mathcal{P}$, predict the possible temperature of their mutated variants in $\hat{\mathcal{P}}$, and measure the ordinal association between prediction and ground truth temperatures of $\hat{\mathcal{P}}$.

**Optimizing Editing Suggestions.** Instead of classifying whether the mutation increases or decreases the thermostability, we introduce an efficient editing algorithm, by optimizing towards an improved thermostability, as depicted below:

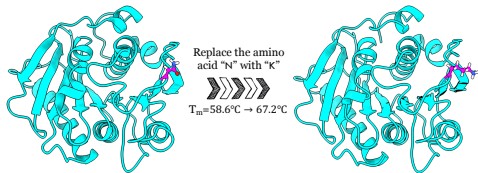

$$\max_{|\mathcal{M}| \leq n} \mathcal{L}(\mathcal{F}(\mathcal{M}(\boldsymbol{x})), y_t), \quad (5)$$

where $\boldsymbol{x}$ is the input, $y_t$ denotes the target label of the class with a higher temperature (*e.g.,* from *Mesophilic* to *Thermophilic*), $\mathcal{M}$ stands for the mutation function whose number of mutated amino acids $|\mathcal{M}|$ is constrained by $n$, $\mathcal{F}$ and $\mathcal{L}$ is our classifier and the objective function. In practice, we set $y_t$ as the highest-temperature class, *Hyperthermophilic*. We set $\mathcal{L}$ in Equation 5 as probability

Figure 4: Visualization of an AI design that thermostabilized a protein. The amino acid "N" was predicted by the MutCompute web server to be mutated to a "K", which was experimentally shown to improve thermostability by $8.6°C$ and was critical for the engineering of FAST-PETase(Lu et al., 2022). ThermoPETase structure (left) (PDB: 6ij6) and the mutated one (right) was obtained via (Waterhouse et al., 2018).

change times the saliency score following PWWS (Ren et al., 2019). See Appendix C.1 for details.

**Experimental Results.** As demonstrated in Table 4, ❶ ESM-1B outperforms TAPE when it is zero-shot transferred to test data. Ours performs the best. We achieve 66.13% accuracy, 13.65% Kendall correlation, and 19.96% Spearman correlation, which is 3.86%, 5.31%, 10.47% relative higher than the ESM-1B baseline. ❷ After finetuning on $\mathcal{P}$, ESM-1B obtains slightly better results than TAPE on both regression and classification metrics. Our model still outperforms both in all the tested cases. ❸ When we do protein editing after fine-tuning, ours achieves the best (54.24% successful rate, 3.49% relative improvements than ESM-1B). It indicates that a better classifier also benefits protein engineering. Visualization of an edited protein is displayed in Figure 4.

## 6 CONCLUSION

Predicting thermostabilizing mutations is a primary goal of most protein engineering campaigns. However, the lack of thermally annotated protein data and effective algorithms has hindered the development of a thermostability prediction model that can generalize across the protein space. This work provides attempts to address both points via a large-scale protein dataset (HotProtein) with *species-specific, lower-bound thermostability annotations* and a novel algorithmic framework designed to tackle the intrinsic challenges of functional domain shifts and data scarcity in thermostability protein engineering. Extensive results validate that our dataset and algorithmic framework provide meaningful improvements over baseline models. Lastly, based on established superior predictors, we search a protein's single-point mutation landscape towards identifying thermostabilizing variants. We will keep updating HotProtein by collecting more sequences, improving our sequence-clustering and filtering, and folding more structures.

ACKNOWLEDGEMENT

This work is in part supported by the NSF AI Institute for Foundations of Machine Learning (IFML), the Defense Threat Reduction Agency (HDTRA1201001), and the Welch Foundation (F-1654). We would like to thank the Reviewers for taking the time and effort necessary to review the manuscript. We sincerely appreciate all valuable comments and suggestions, which helped us to improve the quality of the manuscript.

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

# A   MORE RELATED WORKS

**Protein Engineering.** In general, the factors that make one protein functional and simultaneously thermostable compared to another are complex and unknown for a specific target protein. The literature on computational models for in silico screening of the thermodynamic impact of a mutation is exhaustive. In the development of biopharmaceuticals and biocatalysts, improving the thermodynamic properties of the target protein is a common early goal in most protein engineering campaigns. A protein's thermodynamic stability is commonly represented by its Gibb's free energy and/or melting temperature. The protein engineering community has been designing computational tools to enable in silico screening of mutations for several decades and these tools can be classified into biophysical models and statistical/machine learning-based models. Computational protein engineering models are used to guide the search for optimized sequences. Despite recent progress, these methods have limited utility in reliably ranking sequences, especially at discerning small changes in thermodynamic properties. Although physics-based methods have been shown to reach reasonable accuracy, they are computationally demanding and low-throughput making them intractable to apply to large-scale in silico screening of protein variants.

**Recent Success in AI + Protein Engineering.** Beyond promising performance in simulation experiments, in the last two years, unsupervised structure-based deep learning techniques have also shown their ability to learn from natural protein structures and then generalize to guide the engineering of structurally diverse proteins (Kuhlman & Bradley, 2019; Shroff et al., 2020). For example, Paik et al. (2021) used MutCompute to improve the thermostability of a DNA polymerase in order to enable faster loop-mediated isothermal amplification (LAMP) (Panno et al., 2020) assays to be carried out at $73°C$, a temperature where the commercial counterpart enzyme, BST 2.0, is inactive. Additionally, Lu et al. (2022) engineered two Polyethylene terephthalate (PET) hydrolases (PETase and Cutinase) with MutCompute to improve its thermal and pH stability, resulting in FAST-PETase: a PET hydrolase that can fully degrade post-consumer PET within 48 hours.

**Directed Evolution and Guided Directed Evolution.** The most common strategy to engineer and stabilize a protein is directed evolution. Directed evolution leverages natural evolution where random mutagenesis is synergized with screening to identify variants with improved fitness for the desired phenotype, essentially performing a greedy local search to optimize protein fitness (Arnold, 2018). For directed evolution, it is necessary to design a high-throughput assay that selects mutants enriched for your target phenotype, which is not always possible. Recent progress has been made in combining machine learning approaches with the data collection capability of directed evolution in order to accelerate iterative rounds of directed evolution (Yang et al., 2019a; Wittmann et al., 2021). Here, machine learning is used to ease the experimental screening burden by evaluating proteins in silico. Briefly, machine learning-guided directed evolution (MLDE) works by iteratively training an ML model on a small number of variants ($10^1$-$10^2$) from a combinatorial library and is then used to infer the remaining variance in the library where the variants with the highest predicted fitness are then experimentally evaluated and then added to the training set for the subsequent round. However, the need to consistently generate libraries and expand the training set does not completely alleviate the screening burden. Furthermore, Wittmann et al. (2021) showed empirical evidence that sequence-based embeddings from transformer models can accelerate MLDE and that an MSA transformer (Rao et al., 2021) embedding outperformed sequence embeddings of larger models such as ProtBert-BFD and ESM-1B (Rives et al., 2021).

**Protein Thermostability Dataset.** In protein engineering, improving the stability of a protein is a goal in nearly all protein engineering campaigns and can be a deciding factor in the commercialization of a biocatalyst. Thus, there is plenty of literature on computational algorithms attempting to model this phenotype. Over the last two decades, several algorithms that predict the thermodynamic impact of mutations have been developed (e.g., Park et al., 2016; Schymkowitz et al., 2005; Dehouck et al., 2009; Jokinen et al., 2018; Worth et al., 2011; Cao et al., 2019; Li et al., 2020; Paradis & Schliep, 2019; Krueger & Andrews, 2011; Dehghanpoor et al., 2018) and can be classified as either biophysical models or statistical/machine learning models. Biophysical models utilize amino acid interactions and conformational/rotamer sampling of the protein structure to determine the changes in stability. The primary two biophysical models are Rosetta (Park et al., 2016) and FoldX (Schymkowitz et al., 2005). These methods have shown good performance and utility in several applications but fail to capture narrow changes in stability ($< 1kcal/mol$), which is commonly observed for many mutations. Furthermore, they are computationally demanding and have low

throughput, hindering their application to large-scale in silico screening for identifying stabilizing point mutations (Li et al., 2020; Paradis & Schliep, 2019). Machine learning tools trained to evaluate the impact of a mutation on the stability of a protein are growing in popularity (e.g., Jokinen et al., 2018; Dehouck et al., 2009; Worth et al., 2011; Cao et al., 2019; Li et al., 2020; Paradis & Schliep, 2019; Krueger & Andrews, 2011; Dehghanpoor et al., 2018). Some notable machine learning models include PopMusic (Dehouck et al., 2009), DeepddG (Cao et al., 2019), ThermoNet (Li et al., 2020), SDM (Worth et al., 2011), and mGPfusion (Jokinen et al., 2018). These models are trained in a supervised fashion to predict available experimental data, usually sourced from ProTherm (Jia et al., 2015) or manually curated datasets from the literature, using either evolutionary or structural input features with some models utilizing both types of features. Thus, all use a relatively small training set, with DeepDDG (Cao et al., 2019) having the largest at 5700 manually curated data points, due to the limited amount of experimentally validated point mutations. So far, we have yet to see a stability-predicting algorithm that leverages a language model's learned representation to predict the stability effects of mutations.

**Sequence- and Structured-based Protein Models.** Motivated by self-supervised pre-training in natural language community (*e.g.,* Liu et al., 2019; Yang et al., 2019b), a variety of recent works formulate the protein pre-training tasks as sequence self-supervised learning, *e.g.,* with auto-encoding (Shuai et al., 2021), auto-regressive (*e.g.,* Rives et al., 2019; Meier et al., 2021; Elnaggar et al., 2020; Riesselman et al., 2019), skip-gram language model (Kimothi et al., 2016), mask prediction (Vig et al., 2020; Brandes et al., 2022) or amino acid contrastive learning objectives (Lu et al., 2020), similarity metric learning (Bepler & Berger, 2019; Alley et al., 2019). Although using different architectures (*e.g.,* LSTMs, transformers, graph networks), different pre-trained datasets (*e.g.,* UniRef100 (Suzek et al., 2015), BFD (Steinegger & Söding, 2018), Pfam (El-Gebali et al., 2019)), and different objectives, these models convert input sequences into per amino acid representations. Most recently, thanks to AlphaFoldV2 and its folded dataset (Jumper et al., 2021), ESM-IF (Hsu et al., 2022) proposes to learn the 3D model with inverse folding, which predicts sequences from backbone 3D structures. The input is 3D coordinate information for each residues backbone N, CA, and C atoms (present in every amino acid). Passing the input through the ESM-IF permutation invariant model, the model outputs a probability distribution for amino acid at each residue position.

**Sequence-based Pretraining Methods.** ESM-1b (Rives et al., 2021), MSA Transformer (Rao et al., 2021), and Vig et al. (2020) are several popular pre-trained transformer protein language models, which regard one amino acid as a token. Shuai et al. (2021) trains an antibody language model with antibody data. These works have been applied to simulation experiments. For example, Hie et al. (2022) performs ESM-guided affinity maturation of seven diverse antibodies, screening 20 or fewer variants of each antibody across only two rounds of evolution. Shuai et al. (2021) demonstrates that their model can be applied to generate synthetic libraries that may accelerate the discovery of therapeutic antibody candidates in real experiments.

Tranception (Notin et al., 2022) proposes new architectures for sequence pretraining and offers new test benchmarks. Different from this work, we are interested in pre-training objectives and fine-tuning methods, instead of neural architectures. FLIP (Dallago et al., 2021) sets up benchmarks for fitness landscape inference for proteins, and theorem stability is one part of the benchmark. DeepET (Li et al., 2022a) first pre-trains their convolution network in sequence and then the model is fine-tuned on thermal prediction tasks. Compared to DeepET, which proposes to fine-tune the last layer or do end-to-end fine-tuning, we introduce the sparse and low-rank fine-tuning method. As shown in Table A6, our regularized fine-tuning reaches superior performance, compared to the last-layer or end-to-end fine-tunings.

**Sparse and Low-rank Subspace Fine-tuning.** *Sparsity-aware fine-tuning* is typically leveraged towards the goal of parameter-efficiency (Rebuffi et al., 2017; Houlsby et al., 2019; Li & Liang, 2021; Lester et al., 2021), which only tunes a few of model weights and keep the rest unchanged. Guo et al. (2021) embeds the sparsity into fine-tuning with a differentiable approximation of $\ell_0$ regularization. Chen et al. (2021e) introduces sparse tuning patterns via classical pruning methods (Han et al., 2015). (Chen et al., 2021a) pre-defines the shape of sparsity patterns and learns their combination. Another alternative solution for efficient fine-tuning is to constrain weight updates within a *low-rank subspace* (Hu et al., 2021; Chen et al., 2021e; He et al., 2021). Numerous literature (Yu et al., 2017; Li et al., 2018; Oymak et al., 2019; Gur-Ari et al., 2018) point out the intrinsic low-rank dimensionality of trained over-parameterized models. Wang et al. (2020); Hu et al. (2021) focus on imposing explicit low-rank structures when transferring pre-trained models to diverse downstream

tasks, leading to considerable parameter efficiency. Recently, several pioneering studies (Chen et al., 2021e;a;b) consider combining sparsity and low-rank decomposition for improved efficient training. However, we investigate these two structural priors from a distinctive perspective, *i.e.,* their implicit regularization effects against data scarcity and domain shifts.

## B  MORE METHOD DETAILS FOR STRUCTURE-AWARE PRE-TRAINING

During the pretraining, the language model only learns from context-token pairs, instead of learning a global representation for one sentence (or one protein). Once we target the global representation of one sentence (or one protein), we should learn from a loss function that directly compares one sentence and another (Gao et al., 2021). Similar to the sentence representations of BERT (Devlin et al., 2018), we notice that the protein representations learned by sequence models are not uniformly distributed in the latent space (Gao et al., 2021). Therefore, we adopt contrastive learning in the protein embedding space to distribute the representations more uniformly and let the 3D model inject information into the sequence model. We refer readers to (Gao et al., 2021; Su et al., 2021) for details about motivations.

**Momentum Encoder to Construct Negative Examples.** Following (He et al., 2020), the momentum model $\mathcal{F}^{\text{mom}}$ is a slowly updated model with update rule: $\mathcal{F}^{\text{mom}} \leftarrow \alpha \mathcal{F}^{\text{mom}} + (1 - \alpha)\mathcal{F}$, where $\alpha \in [0, 1)$ is the momentum coefficient. During the optimization of Equation 1, to save the computation budget for negative examples, we adopt the memory bank (He et al., 2020) to construct enough negative examples. The Neg set is taken to be a queue of $\mathcal{F}'(\boldsymbol{x}_j)$ representations from previous mini-batches.

## C  MORE IMPLEMENTATION DETAILS

**Computing resources.**  Experiments use `Tesla V100-SXM2-32GB` GPUs as computing resources. Each experiment can be run with a single `V100` GPU.

### C.1  EXPERIMENTAL DETAILS ABOUT PROTEIN EDITING

**Search Space.** Based on our fine-tuned classifiers, we generate 20 possible single point mutations at each position ($i$) in a natural protein sequence $p^{(i)}$ and then evaluate whether the resulted $\hat{p}_k^{(i)}$ ($k \in \{1, \cdots, 20\}$) successfully leads to improved thermostability. The corresponding successful rate is used as our evaluation metric.

**Optimization Method.** We adopt a recently-proposed well-known method, PWWS (Ren et al., 2019). PWWS replaces tokens base on a ranking score $H(x, x_i^*) = \phi(S(x))_i \Delta P_i^*$, where $x$ is the input sequence, $x_i^*$ stands for replacing the amino acid at index $i$, $\Delta P_i^*$ denotes the change in classification probability after replacing with $x_i^*$, and $\phi(S(x))_i$ is the saliency score.

**Dataset Descriptions.** Previously, numerous benchmarks have been proposed for thermostability or $\Delta\Delta$G prediction. Some of them use Rosetta (Rohl et al., 2004) predicted score, (Frenz et al., 2020) and others create datasets, *e.g.*, ProThermDB (Nikam et al., 2021) based on experiment records. The curation of large high-quality thermostability datasets is still ongoing, and protein stability experiments are time-consuming and expensive (Stourac et al., 2021). We set up our benchmark using FireProtDB (Stourac et al., 2021), which is a superset of published experiments records and is the most up-to-date dataset to our knowledge. We download the latest version of FireProtDB, clean the duplicates, and extract the $\Delta Tm$ values. We retain all the values with $\|\Delta Tm\| \geq 1$ to make sure that the temperature change is large enough and not random noise. Finally, we get 961 data instances with $\|\Delta Tm\| > 1$.

**Benchmark Quality.** Rosetta is an academic framework for computational modeling and analysis of protein structures. We utilize the Rosetta Cartesian $\Delta\Delta$G application (Park et al., 2016) to assess how well-established biophysical computational tools can recapitulate the thermostability dataset - FireProtDB. The Cartesian $\Delta\Delta$G application calculates the change in the folding energy of a mutation. First, we relax the crystal structures of the proteins in FireProtDB with an unconstrained FastRelax as recommended in Leman et al. (2020), which allows the backbone and side-chain atoms to move slightly to be better accommodated into the chosen Rosetta score, "ref2015_cart". Cartesian

Table A5: We report the correlation coefficient between Rosetta $\Delta\Delta$G and FireProtDB $\Delta$T$_m$, $\Delta\Delta$G.

| FireProtDB | Rosetta | Data Size | Spearman (%) | Pearson (%) |
|---|---|---|---|---|
| $\Delta\Delta$G | with backbone relaxation $\Delta\Delta$G | 3,399 | 44.93 | 21.36 |
| $\Delta\Delta$G | w/o. backbone relaxation $\Delta\Delta$G | 3,248 | 28.59 | 3.31 |
| $\Delta$T$_m$ | with backbone relaxation $\Delta\Delta$G | 981 | -35.91 | -3.98 |
| $\Delta$T$_m$ | w/o. backbone relaxation $\Delta\Delta$G | 1,018 | -13.54 | 0.26 |

$\Delta\Delta$G then mutates the residues as specified in the database, packs side-chain conformations, and does gradient-based minimization of the atomic coordinates. Using the values from the Rosetta score, we can then calculate the $\Delta\Delta$G of mutation. Any mutations that failed during the calculation for any reason were discarded.

We calculate the correlation between simulation $\Delta\Delta$G and experimental $\Delta\Delta$G in Table A5. Allowing backbone relaxation during structure generation improves the Spearman $\Delta\Delta$G to 44.93 from 28.59 and the Spearman $\Delta$T$_m$ to $-35.91$ from $-13.54$. Compared to Table 2, we observe that our proposal offers better candidate mutations in terms of the spearman correlation coefficient.

**Training Settings.** For zero-shot transfer, we directly apply the models trained on HP-S to $\hat{\mathcal{P}}$. For fine-tuning, we train all the models with AdamW optimizer, $5 \times 10^{-2}$ weight decay, and the OneCycle learning rate schedule. TAPE models are trained with batch size 8, learning rate $10^{-3}$ and 10 epochs, while ESM models are trained with batch size 16, learning rate $10^{-3}$ and 20 epochs. For Deep Editing, we use models trained on $\mathcal{P}$. When we finetune ESM models, we re-initialize the final-layer linear head.

# D  MORE EXPERIMENTAL RESULTS

Table A6: Head Tuning versus Full Tuning of ESM-1B on thermostability classification and regression. 95% confidence interval are computed via the 10-fold evaluation on HP-S$^2$C2/C5 and 3 replicates on HP-S.

| ESM-1B | HP-S$^2$C2 | | HP-S$^2$C5 | | HP-S | |
|---|---|---|---|---|---|---|
| | Accuracy | Spearman | Accuracy | Spearman | Accuracy | Spearman |
| Head Tuning | **91.19**±**0.47** | **0.890**±**0.018** | **83.26**±**1.54** | **0.712**±**0.043** | **69.50**±**0.16** | **0.809**±**0.001** |
| Full Tuning | 88.75±1.29 | 0.797±0.029 | 77.97±2.46 | 0.503±0.058 | 65.88±0.89 | 0.615±0.050 |

**Head Tuning versus Full Tuning of ESM-1B.** In Table A6, only tuning the head of ESM-1B consistently achieves better performance than tuning the whole ESM-1B across all three datasets. A potential explanation is that the fully tuned ESM-1B tends to overfit due to the relatively small size of HotProtein (182K), where the pre-training dataset has around 22M protein sequences.

Table A7: Hyperparameter tuning of the Mixup augmentation. ESM-1B equipped with our proposals is adopted with HP-S$^2$C2 and HP-S$^2$C5.

| Settings/Acc. (%) | HP-S$^2$C2 | HP-S$^2$C5 |
|---|---|---|
| $\lambda \sim \mathcal{B}(0.2, 0.1)$ | 0.8547 | 0.8638 |
| $\lambda \sim \mathcal{B}(0.1, 0.2)$ | 0.8600 | 0.8527 |
| $\lambda \sim \mathcal{B}(0.2, 0.5)$ | 0.8675 | 0.8596 |

Table A8: Evaluation results of our proposals with ESM-1B on the Meltome Atlas benchmark.

| Methods | Spearman | Pearson |
|---|---|---|
| ESM-1B + Ours | **0.4560** | **0.6866** |
| ESM-1B | 0.3874 | 0.5331 |
| TAPE | 0.3076 | 0.3132 |

**Hyperparameter Tuning.** In general, we find that our method is not very sensitive to hyperparameter tuning to yield good results. Specifically: (1) Many hyper-parameters were left at default values. For example, for SAP, we used the hyper-parameters from He et al. (2020) without optimizing. (2) For the other hyper-parameters (e.g. batch size, learning rate), we used standard 10-fold cross-validation for selection. (3) Our algorithm is not sensitive to the choice of hyper-parameters, as shown in Table 2, our FST achieves considerable improvements for various hyper-parameter

choices. (4) We provide more ablations for Mixup augmentation in Table A7, where $\lambda$ is driven from beta distributions $\mathcal{B}$.

**Our Proposal on Other Benchmarks.** To demonstrate the generalization ability of our methods, we further perform the zero-shot transferring on Meltome Atlas (we train the model on HotProtein and directly evaluate the model's performance on 8K Meltome Atlas data). We notice that our method still achieves better results in Table A8.

Table A9: Comparison with more baselines of MLP and ESM-1B without any pre-training.

| Settings/Acc. (%) | HP-S$^2$C2 | HP-S$^2$C5 |
|---|---|---|
| ESM-1B w.o. pre-training | 0.7808 | 0.6652 |
| ESM-1B w. pre-training | **0.9119** | **0.8326** |
| MLP | 0.6931 | 0.6378 |

Table A10: More approaches to inject the structure information with ESM-1B.

| Settings/Acc. (%) | HP-S$^2$C2 | HP-S$^2$C5 |
|---|---|---|
| Ours | **0.9236** | **0.8625** |
| Add | 0.8873 | 0.8093 |
| Concat | 0.9001 | 0.8186 |

**More Baselines: Small Neural Networks and ESM-1B without Pre-training.** We conduct extra experiments with 1) ESM-1B without the pretraining weights (with 2 times larger number of training iterations than an MLP's) and 2) a small neural network, 3-layer MLP (256 Dimensional Embedding Layer $\rightarrow$ 3 Layer MLP $\rightarrow$ Average Pool) in Table A9. These two approaches achieve much worse results than ESM-1B baseline.

**Other Approaches to Inject Structure Information.** As shown in Table A10, we notice that directly concatenating or adding the final-layer representations of ESM-IF (structure information) and ESM-1B (sequence information) comes to slightly worse results.

**More Comparisons: End-to-end or Freezing Some Layers Tuning.** We provide additional results with ESM-1B on HP-S$^2$C2 and HP-S$^2$C5. "First 1 / 3 Layers" indicates the layers close to the input. To achieve better results, we tune the backbone learning rate with 10-fold cross-validation for these numbers. Table A11 tells us that only tuning the head leads to superior performance.

**Evaluation on Balanced Test Sets.** We perform extra evaluations on manually balanced test sets. Results are summarized in Table A12, where our methods consistently show superior performance.

Table A11: Comparison with the end-to-end and partially frozen tuning with ESM-1B on HP-S$^2$C2/5.

| Settings/Acc. (%) | HP-S$^2$C2 | HP-S$^2$C5 |
|---|---|---|
| End-to-End | 0.8875 | 0.7797 |
| Freeze First 1 / 3 Layers | 0.8927 | 0.8018 |
| Freeze First 2 / 3 Layers | 0.9011 | 0.8195 |
| Tuning Heads | 0.9119 | 0.8326 |
| Tuning Heads + Ours | **0.9236** | **0.8625** |

Table A12: Evaluation on manually balanced datasets.

| Settings/Acc. (%) | HP-S$^2$C2 | HP-S$^2$C5 | HP-S |
|---|---|---|---|
| 3D GCN | 0.7633 | 0.5004 | - |
| TAPE | 0.8117 | 0.5535 | 0.6370 |
| ESM-IF1 | 0.7752 | 0.6032 | - |
| ESM-1B | 0.8605 | 0.8029 | 0.6980 |
| ESM-1B + Ours | **0.9308** | **0.8294** | **0.7517** |

Table A13: Ablation on the feature aggregation methods with ESM-1B on HP-S$^2$C2 and HP-S$^2$C5.

| Settings/Acc. (%) | HP-S$^2$C2 | HP-S$^2$C5 |
|---|---|---|
| Average Pooling | **0.9119** | **0.8326** |
| Max Pooling | 0.8819 | 0.3750 |
| No Pooling | 0.9084 | 0.7808 |

Table A14: Comparisons with the $\ell_1$ and $\ell_2$ regularized tuning for ESM-1B on HP-S$^2$C5.

| Settings/Acc. (%) | HP-S$^2$C5 |
|---|---|
| ESM-1B | **0.8326** |
| ESM-1B + $\ell_2$ (Ridge Regression) | 0.7375 |
| ESM-1B + $\ell_1$ (Lasso) | 0.3808 |

**Ablation on the Feature Aggregation Methods.** In Table A13, we conduct ablation studies on the feature aggregation methods with ESM-1B on HP-S$^2$C2 and HP-S$^2$C5, including average pooling, max pooling, and no pooling. The results are summarized in the below table. We observe that the average pooling outperforms other aggregation options.

**Comparison to $\ell_2$ (Ridge Regression) and $\ell_1$ (Lasso) Regularization.** We conduct comparisons to $\ell_2$ (ridge regression) and $\ell_1$ (Lasso) regularization with ESM-1B on HP-S$^2$C5. Results are in Table A14. We observe that additional regularizers may lead to performance degradation.

**Additional Results on Editing** In Table 4, we use to 5-class classifier trained on HP-S, and we provide additional results with 2-class classifier trained on HP-SC2. We get zero-shot accuracy 65.66%, precision 42.77% and successful rate 42.14%. After fine-tuning, we get accuracy 70.82%± 0.32%, precision 48.31 ± 0.29% and successful rate 53.05 ± 0.26%. The result is slightly worse than the 5-class classifier results.

## E    MORE DATASET DETAILS

**Data Collection and Process.** All the data in the NCBI bioproject is accumulated from all scientists who publish data that makes it into the NCBI database. Thus, there is a lot of duplication and variation in data entry. Due to the vast number of organisms and multiple strains for organisms, we removed duplicates by taking the first observation in the NCBI bioproject that had a consistent quantitative optimal growth temperature and temperature classification.

We remove all sequences that are greater than 1500 amino acids, since most proteins of interest for engineering fall within this range and it greatly simplifies clustering. During the clustering, we cluster across all organisms in the same temperature bin in order to remove redundancy across organisms and kept representative sequences from each cluster in order to not bias sampling to a specific organism.

There are numerous organism-specific idiosyncrasies present (such as sequence homology between organisms due to their evolutionary relationship) in each organism proteome that has nothing to do with thermostability and instead their unique adaptation to their environment. However, there is no clear-cut way to identify these features/sequences and remove them. Therefore, we expect there to be numerous organism-specific idiosyncrasies present (such as sequence homology between organisms due to their evolutionary relationship) in each organism proteome that has nothing to do with thermostability and instead their unique adaptation to their environment. In practice, we do not further filter the data to reduce evolutionary differences. Instead, we cluster proteins based on sequence similarity across all organisms within a temperature bin and select the representative sequence of each cluster to avoid biasing our dataset to a particular organism.

**Domain Shift.** For domain shifts, it includes two perspectives: ❶ *Between HotProtein and FireprotDB*. Annotating a proteome with the optimal growth temperature of the organism provides us a lower bound soft label for training on nearly 200K sequences (i.e., HotProtein). However, we expect to observe a domain shift when fine-tuning on experimentally curated stability datasets (i.e., FireprotDB) since these labels accurately represent a protein variant thermodynamic properties, unlike our coarse, optimal growth temperature label. We use the classic tool, i.e., **cd-hit** [5], to compute the protein sequence similarity (1) between FireprotDB and Hotprotein; (2) within FireprotDB. We find the similarity between FireportDB and Hotprotein (0.1928) is much lower than the one within FireprotDB (0.2504). It suggests the substantial domain shifts, echoed with our paper's descriptions. ❷ *Within HotProtein*. HotProtein covers 230 in different species where proteins have different functionality and structures. For example, we expect to observe domain shifts between proteins sampled from the eukaryotic species (primarily unicellular fungal) and the prokaryotic organisms.

**Data Scarcity.** For the data scarcity, we measure our framework on two kinds of data-limited scenarios: (1) We have evaluated our proposals on subsampled HP-S$^2$C2 and HP-S$^2$C5, and demonstrated their effectiveness. HP-S$^2$C2 and HP-S$^2$C5 only have around 2K sequences ($\frac{2}{183}$ of the whole Hot-Protein dataset). (2) We further examine our approaches on FireprotDB which is a manually curated database of the protein stability data for single-point mutants. It only contains 0.2K natural protein amino acid sequences which serve as the training set in our case. Table 4 shows that our methods lead to consistent performance improvements.

**An Organism Lives is Correlates with the Thermostability of the Protein Sequences of that Organism.** The Meltome Atlas provides thermal proteome profiling (TPP) for 13 organisms (Jarzab et al., 2020). Although TPP is not the melting temperature of the protein since it is not measured

---

[5] http://weizhong-lab.ucsd.edu/cd-hit/

with purified proteins and requires the protein to become insoluble upon denaturation (proteins can remain soluble even after denaturation), it provides empirical evidence that for mesophilic, and thermophilic prokaryotic organisms, the optimal growth temperature of an organism is very close to the lower bound of the thermal stability of that organism. This is not the case for eukaryotic and psychrophilic prokaryotes where the Meltome Atlas shows a ¿10C gap between the optimal growth temperature and the lower bound of protein stability in these organisms. Thus, given the available proteome experimental data, we assume that using the optimal growth condition for prokaryotic mesophiles and thermophiles provides an accurate lower bound melting temperature for their respective proteome. Furthermore, although our method utilizes the lower bound of a proteome's thermal stability as a label, we demonstrate that it provides a sufficient learning signal to improve performance on the manually-curated experimental dataset: FireProtDB.

## F   DISCUSSION OF BROADER IMPACT

This research aims to predict the thermal stability of proteins from the sequence and structural data and predict thermostabilizing mutation designs. Improving a protein's robustness to thermal challenges can often be the deciding factor for the commercialization of a biocatalyst. Furthermore, the ability to rapidly thermostabilize a protein will open the door for the engineering of a broader range of biotechnologically relevant enzymes and therapeutics, which can have a profound impact on the chemical, agricultural, food, and pharmaceutical industries. It is well documented that functional residues tend to be destabilizing residues and engineering function into a protein can quickly destabilize the protein. Thus, it is common to first improve a protein's stability prior to introducing destabilizing mutations. Our work attempts to accelerate the initial stabilization of a target protein to enable downstream functional protein engineering.

| ESM-1B | HP−S$^2$C2 | | HP−S$^2$C5 | | HP−S | |
|---|---|---|---|---|---|---|
| | Accuracy | Spearman | Accuracy | Spearman | Accuracy | Spearman |
| Full Tuning | 88.75±1.29 | 0.797±0.029 | 77.97±2.46 | 0.503±0.058 | 65.88±0.89 | 0.615±0.050 |
| Head Tuning | 91.19±0.47 | 0.890±0.018 | 83.26±1.54 | 0.712±0.043 | 69.50±0.16 | 0.809±0.001 |
| Ours | **91.91**±**0.64** | **0.892**±**0.011** | **86.08**±**1.33** | **0.747**±**0.026** | **73.09**±**0.10** | **0.823**±**0.001** |

