# OpenReview forum: "HotProtein: A Novel Framework for Protein Thermostability Prediction and Editing"
_ICLR.cc/2023/Conference — ICLR 2023 poster_

### Official Review · Reviewer_jaKq · 2022-10-24

**Confidence:** 4
**Correctness:** 3
**Technical Novelty And Significance:** 3
**Empirical Novelty And Significance:** 3
**Recommendation:** 5

**Clarity, Quality, Novelty And Reproducibility:**

The paper is clearly written, and the experiments are comprehensive. However, it's hard to say whether the model would really be useful in resolving biological problems.

**Strength And Weaknesses:**

Strengths:
1.	The amount of data for thermostability-related tasks is greatly expanded.
2.	The writing is generally clear, and easy to read.
3.	Editing proteins to improve thermostability is interesting, biological experiments can be added in the future.
4.	The experiments on the model are relatively comprehensive.
Weakness:
1.	Their pretraining module simply combines the sequence pretrained model and the 3-D structure-aware pre-trained model. No novelty on the loss function can be viewed.
2.	Two reasons mentioned for using factorized sparse tuning for this prediction task are not convincing. The sentence ‘while COVID-19 related proteins usually contain more than 1,000 amino acids’ has no strong connection with the studied problem.
3.	From the result, the framework seems just introduce some training strategies based on ESM-1B, and there is no design in biology for the thermostability prediction task. Besides, no significant improvement in results is observed.
4.	The description of biology in this paper is superficial, and the study lacks interpretability analysis.


**Summary Of The Paper:**

This paper presents a framework for protein thermostability prediction. A large-scale protein dataset with organism-level temperature annotations is curated, and one pretraining and one tuning module are proposed for prediction.
The main contributions are:
1.	A dataset of protein sequences and folded 3D structures data with temperature annotations.
2.	Introducing a protein structure-aware pre-training module for protein-related tasks.
3.	Proposing a factorized sparse tuning module for thermostability prediction.


**Summary Of The Review:**

The paper is clearly written, and the experiments are comprehensive. However, it's hard to say whether the model would really be useful in resolving biological problems.

---

> ### Author Response · Authors · 2022-11-19
> **Response to Reviewer jaKq [Cons 1-5]**
>
> We are glad that reviewer jaKq appreciates the contribution of our cleaned dataset and comments our writing as “generally clear and easy to read”, our editing proposal as “interesting”, and our experiments as “comprehensive”. To address reviewer jaKq’s questions, we provide pointwise responses below.
>
> **[Cons 1. No Novelty on the Pre-training Loss.]**  We are the first to learn better feature extractors using contrastive learning upon 3D and sequence data pairs of protein. Once we target the global representation of one sentence (or one protein), we should learn from a loss function that directly compares one sentence and another (Gao et al., 2021). Similar to the sentence representations of BERT (Devlin et al., 2018), we notice that the protein representations learned by sequence models are not uniformly distributed in the latent space (Gao et al., 2021). Therefore, we enhance the protein’s representation power using contrastive loss.
>
> **[Cons 2. Reasons for Using Factorized Sparse Tuning are not Convincing.]** We politely point out that our factorized sparse fine-tuning is well-motivated. The reasons lie in the following aspects:
>
> *<Two Challenges Exist.>*
> (1) Data Scarcity. In real-world applications, it is extremely challenging or even infeasible to collect a sufficient large-scale (engineered) protein dataset, due to the massive time and resource cost of transformation, protein expression, and purification. For example, the FireProtDB dataset used in our paper only contains ~200 wild-type protein sequences.
> (2) Data Distribution Shifts. Proteins with different (sometimes even the same (Xia, 2021)) functionalities have distinctive and idiographic structures, sharing few common characteristics. For instance, they may differ in sequence length, amino acid distributions, protein structures, and so on. The mentioned COVID-19 related protein is one example. A more detailed explanation of data distribution shifts is provided in Section E of our appendix.
>
> *<These Challenges Fail Naive Fune-tuning.>* As shown in Table A6, fine-tuning the full backbone of ESM-1B suffers from degraded performance (i.e., worse than only fine-tuning the linear head), due to the aforementioned challenges in protein data. Meanwhile, as demonstrated in Tables 1, 2, and 4, our factorized sparse tuning consistently surpasses the baseline of only fine-tuning the head across different scenarios. It is because our enforced critical low-rank and sparse structures in the model’s update space, regularize the fine-tuning and alleviate the negative effects of the above two data-level challenges.
>
>
> **[Cons 3. No Design in Biology for the Thermostability Prediction.]** The biological intellectual contribution lies in the HotProtein dataset generation leveraged for pre-training. We specifically generate bins of protein sequences based on the optimal growth temperature of organisms and then train classifiers and regressors to extract underlying features that enable discernment of the organism's optimal growth temperature. We then take this pre-trained model and zero-shot and fine-tune it on FireProtDB.
>
> We politely point out that our algorithms are well-motivated to resolve the challenges in modeling protein data (Please kindly refer to **Cons 2**). Yes, our algorithmic framework is potentially generalizable to more protein phenotypes. We see this as a merit rather than a shortcoming, as also acknowledged by reviewer Uvn5.
>
> **[Cons 4. Improvements are Not Significant.]** We respectfully point out that our improvements are consistent and sometimes statistically significant (non-overlapped 95% confidence intervals).
> As presented in Tables 1, 2, 4, and A8, for all evaluations of protein thermostability prediction and mutation suggestion on HotProtein, FireProtDB, and Meltome Atlas benchmarks, our proposals bring consistent performance improvements.
> With respect to the 95% confidence interval, our approaches are significantly better than all baselines in Table 1 in terms of accuracy, and in Table 4.
>
> **[Cons 5. Lack Biology Descriptions and Interpretability Analysis.]** Please kindly refer to **Cons 3** for biology descriptions. The underlying model has 650M parameters and interpreting what these weights learn is a significant undertaking, which we believe is beyond our scope. We will work on introspecting our fine-tuned model in our future research.

---

> ### Author Response · Authors · 2022-11-26
> **Response to Reviewer jaKq**
>
> Dear Reviewer **jaKq**,
>
> We thank reviewer **jaKq** time for the review and for constructive comments. We really hope to have a further discussion with the reviewer **jaKq** to see if our response solves the concerns.
>
> In our response, we have (1) clarified the motivations of our algorithms; (2) explained the biology designs.
>
> We genuinely hope reviewer **jaKq** could kindly check our response. Thank you!
>
> Best wishes,
>
> Authors

---

### Official Review · Reviewer_un7C · 2022-10-24

**Confidence:** 4
**Correctness:** 2
**Technical Novelty And Significance:** 2
**Empirical Novelty And Significance:** 2
**Recommendation:** 3

**Clarity, Quality, Novelty And Reproducibility:**

Whilst I quite like the model framework. I cannot evaluate it's advantage for the declared task as baselines are missing.

**Strength And Weaknesses:**

A fundamental weakness of this paper is similar conceptually to many papers submitted to ML conferences on biological data -that is the way the train-test splits are made which is 10 fold cross validation. What this means is that they have ‘data leakage’ relative to what it seems they claim to predict -- maximally they show if you had 90% of the data with related  sequences in the split, you could predict the missing 10%. Not a very useful predictor Now imagine a test split that removes a family of sequences that has no sequence relation to training data - then one could say a load more about the ability of the model to generalize

With thousands of proteins many of which are homologous i.e. similar  and very few classes in theri stes (2 or 5 )  - it's hard to see why one even needs a language model when a simple sequences similarity search as a baseline could do the same. Have the authors tested a simple baseline?

The sentence in the abstract “We present HotProtein, a large-scale protein dataset with growth temperature annotations of thermostability..” strongly suggests  they are reporting this dataset here for the first time - when in fact it was published 2 years ago and is publicly available. Since they do in fact clear the data for more facile use, the statement should make that distinction clear.
Related work seems to be missing - especially where methods have collected datasets but also run models to predict mutation effects - some of which are thermostablity related ; since some of these are even unsupervised, using only evolutionary sequence alignments - minimally a  discussion of this body of work is warranted eg - see ProteinGym ( Tranception ICML 2022)  plus older papers that use VAEs ( Riesselman Nature Methods 2018)  and even Potts Models (eg Hopf et al 2017 Nature Biotech) plus MAVEsdb.  In addition FLIP ICML 2021 and 2022- Learning deep representations of enzyme thermal adaptation. Why no comparison  to  ESM1v ( Meier 2021), Tranception ( Notin 2022), FLIP 202, DeepET (2022) ?


Minor
Explain why theory are not using the Meltome measurement as their thermostability label
Although it’s fine to explore whether using structural information improves existing approaches , the sentence on page 4  is misleading for different reasons -- “These models achieve considerable improvements on amino acid prediction tasks, e.g., contact prediction, mask prediction (Brandes et al., 2022), mutational effect prediction (Meier et al., 2021).” Compared to what? Meier’s results that the authors reference are great  BUT are only very marginally better than a VAE ( DeepSequence 2018 Nature Methods) and not as good as Tranception ( Notion et al,  ICML 2022) and are zero shot!! Using only sequence info.
The structure barely improves the results - why do they think that is the case?


**Summary Of The Paper:**

The authors present a new model for predicting protein thermostability based on a published dataset (Jarzab et al Nature Methods 2020) and create three derived different datasets annotating each of proteins with a class name that reflects the temperature that the organism lives at  (retrieved from NCBI). The task then becomes the ability to predict the correct class. They present two algorithmic ‘advances’: structure-aware pretraining and factorized sparse tuning  improve thermostability prediction. After evaluating compared other methods for prediction they also propose a method to generate sequences with increased thermostabilty.


**Summary Of The Review:**

For the reasons given in the Weaknesses section I don't believe this paper achieves what the authors would indicate are their goals - thermostability prediction and design.  The fact that they do not test generalizability, limits the application of this work to where there are very large datasets  of experimental results on related proteins from environmentally related organisms. I would suggest that the authors rethink their evaluations, test with much more naive baselines.

---

> ### Author Response · Authors · 2022-11-19
> **Response to Reviewer un7C [Cons 1-3]**
>
> Thank reviewer un7C’s valuable time and comments. To address reviewer un7C’s questions, we provide pointwise responses below.
>
> **[Cons 1. Model Performance Validation.]** We respectfully point out that we do test the generalizability of our proposals. The reasons lie in the following aspect.
>
> *<Our Focus of Protein Editing and The Sequence Similarity Between Hotprotein and FireProtDB.>* It is noteworthy to mention that our focus or final goal is to provide positive single-point mutations that improve protein thermostability. To validate the effectiveness of our proposals, transfer studies (including zero-shot transfer) are conducted from the temperature prediction on HotProtein to amino acid mutation suggestions on FireProtDB. Results in Table 4 show the superiority of our methods. Note that there exists a substantial discrepancy between them in terms of both sequence similarity and task objectives.
> As specified in Section E of our appendix, the sequence similarities between HotProtein and FireProtDB are low. To be specific, we use the classic tool, i.e., CD-Hit, to compute the protein sequence similarity (1) between FireprotDB and Hotprotein; (2) within FireprotDB. We find the similarity between FireportDB and Hotprotein (0.1928) is much lower than the one within FireprotDB (0.2504). It suggests the substantial domain shift between Hotprotein and FireProtDB.
> Moreover, the task discrepancy between temperature prediction on HotProtein and editing suggestion on FrieProtDB is significant, even for the same protein sequences. The former only models the temperature of target proteins, and has no knowledge about the temperature change given single-point mutations. However, the latter focuses on modeling the temperature changes from single-point mutation.
>
> *<Sequence Similarity in Hotprotein.>* As indicated in Section 3, during the preprocess of HotProtein, we utilize CD-Hit (http://weizhong-lab.ucsd.edu/cd-hit/) to cluster protein sequences across organisms within a “TemperatureRange” class at a sequence similarity threshold of 50%. In each cluster, we only keep one protein whose sequence length is between 200~500. Therefore, HotProtein has already removed many “redundant” sample sequences.
>
> *<Additional Evaluation on Hotprotein with New Train-Test Splits.>* To further address reviewer un7C’s concern, we conduct extra evaluations with new train-test splits beyond the cross-validation. Specifically, we randomly sample 10% of proteins as a test set and treat the rest 90% of proteins as the training set. Note that any samples from the new test set have a sequence similarity below 50% to training samples. Under this new evaluation scheme, our methods obtain 89.42% accuracy and 82.04% spearman correlation on HP-S, i.e., 3.17% accuracy boosts and 6.65% correlation gains compared to the ESM-1B baseline. Observations and conclusions remain the same.
>
> **[Cons 2. The Simple Sequences Similarity Search Baseline.]** The simple sequences similarity search can not work, which is mainly because we have already removed the “redundant” sample sequences if their similarity is beyond 50% (Please kindly refer to **Cons 1**).
> Specifically, to further convince reviewer un7C, we implement the KNN approach based on the sequence similarity. It can only achieve **21.15%** accuracy on HP-S2C5, while ours obtain **86.25%** accuracy as shown in Table 1. The significant performance margin evidence that the naive baseline does not work and our proposals are effective.
>
> **[Cons 3. A More Clear Statement in Abstract.]** We would like to thank the reviewer for pointing this out. Our intention was not to claim that we are the first to publish this raw data but rather to provide the community with a pre-processed, sequence-balanced subset of the raw temperature data. Similar to how ProteinGym provides a cleaned-up pre-processed dataset of all publicly available DMS datasets. Therefore, we modify the sentence in the abstract as “We present HotProtein, a large-scale protein dataset **curated from publicly available** growth temperature annotations of thermostability..”.

---

> ### Author Response · Authors · 2022-11-19
> **Response to Reviewer un7C [Cons 4-7]**
>
> **[Cons 4. Missing Related Works.]** We add discussions about these related works in our revision. Tranception proposes new architectures for sequence pretraining and offers new test benchmarks. Different from this work, we are interested in general pre-training objectives and fine-tuning methods, instead of neural architectures. FLIP sets up benchmarks for fitness landscape inference for proteins, and theorem stability is one part of the benchmark. DeepET, as a concurrent work, first pre-trains their convolution network in sequence, and then the model is fine-tuned on thermal prediction tasks. Compared to DeepET fine-tuning, which proposes to fine-tune the last layer or do end-to-end fine-tuning, we introduce the sparse and low-rank fine-tuning method. As shown in Table A6, our regularized fine-tuning reaches superior performance, compared to the last-layer or end-to-end fine-tunings.
>
>
> **[Cons 5. Comparison with ESM1v, Tranception, FLIP, DeepET?]**
>
> *<Comparison to ESM-1V and Tranception.>* We carry out additional comparisons with ESM-1V and Tranception, as presented in Table R2. Experiments are conducted on FireProtDB using zero-shot-inference. The absolute values of Spearman and Pearson correlations are reported. We see our method outperforms these baselines by a significant performance margin.
>
>
> Table R2. Performance comparisons on the protein of P06654 in FireProtDB
>
> | Method | \|Spearman\|
> | :----: | :----: |
> | ESM-1v | 0.1573 |
> | Tranception | 0.1724 |
> | Ours | 0.1996 |
>
> Meantime, FLIP (https://openreview.net/forum?id=p2dMLEwL8tF) we found is a benchmark rather than an algorithm. If reviewer un7C could kindly point out the specific method reference of FLIP, we are more than happy to compare with it.
>
> *<Comparison to DeepET.>* (1) We politely point out that it is a **concurrent** work. It was first published on Oct. 19, 2022, and ours was submitted to ICLR 23 on Sep. 21, 2022. (2) Meanwhile, we have conducted comparisons in Table A6. Compared to DeeET fine-tuning which proposes to fine-tune the last layer or do end-to-end fine-tuning, our regularized fine-tuning reaches superior performance.
>
>
>
> **[Cons 6. Why Not Use the Meltome Measurement as the Label?]** Actually, we have also benchmarked on Meltome Atlas as demonstrated in Table A8, where the observations are consistent with the ones in our main text. Meanwhile, we choose HotProtein in the main paper, because:
> HotProtein has a wider range of species (**230**) and temperature ranges (**-20 Celsius to 120 Celsius**), while Meltome Atlas has **13** species and covers 30 Celsius to 90 Celsius.
> HotProtein has more protein sequences (**182K**), while Meltome Atlas has (**48K**).
> HotProtein has additional structural data for part of protein samples.
>
> **[Cons 7. Misleading Sentence on Page 4 -- “These models achieve considerable improvements on amino acid prediction tasks, e.g., contact prediction, mask prediction (Brandes et al., 2022), mutational effect prediction (Meier et al., 2021)”.]**
>
> Thanks. We clarify this sentence below:
> We rephrase this sentence by providing the specific comparison targets, as “These pre-trained models achieve considerable improvements compared to traditional computation or sequence alignment methods (Schymkowitz et al., 2005; Montanucci et al., 2019; Chen et al., 2020) on amino acid prediction tasks, e.g., contact prediction, mask prediction (Brandes et al., 2022), mutational effect prediction (Meier et al., 2021; Notin et al., 2022; Li et al., 2022a).” The change has been included in our revision.
> The discussions about Tranception and DeepET have been included. Please kindly refer to **Cons 4** and **Cons 5**.
> We also politely point out that the existing works like Transception, do NOT claim the 3D structure information is useless.

---

> ### Author Response · Authors · 2022-11-26
> **Reponse to Reviewer un7C**
>
> Dear Reviewer **un7C**,
>
> We thank reviewer **un7C** time for the review and constructive comments. We really hope to have a further discussion with the reviewer **un7C** to see if our response solves the concerns.
>
> In our response, we have (1) clarified the setting of performance validation and provided extra evaluations; (2) conducted additional comparisons with the simple baseline and other mentioned literature.
>
> We genuinely hope reviewer **un7C** could kindly check our response. Thanks!
>
> Best wishes,
>
> Authors

---

### Official Review · Reviewer_Uvn5 · 2022-10-27

**Confidence:** 5
**Correctness:** 4
**Technical Novelty And Significance:** 3
**Empirical Novelty And Significance:** 4
**Recommendation:** 8

**Clarity, Quality, Novelty And Reproducibility:**

**Clarity**: Overall the paper is well written and easy to follow, with an extensive related work section. Some sections could be made a bit clearer, eg., section 4.1 (as discussed above).

**Quality**: Very thorough sets of experiments and ablations. Besides some concerns on validation as mentioned above, the experimental design is well-thought through.

**Novelty**:
- Several meaningful / novel contributions, in particular the HotProtein dataset and the structure-aware pre-training. The former is based on existing datasets (NCBI bioproject and Uniprot), but required curation to create the final dataset, and could be a valuable resource for the community. The latter is relatively general and its application could extend beyond the thermostability use case discussed in this work.
- The fine-tuning approach and optimization approach for protein editing are borrowed from other works (respectively [3] and [4]). While authors properly cite the corresponding works, the language used in section 1 when listing contributions is a bit misleading (eg., “we propose …”, “we built …”). Would recommend that authors use a more accurate language there to properly reflect their contributions (eg., “we adapt…”)

**Reproducibility**: Main information needed for reproduction are provided. Authors mention that code and data will be made public in the abstract.

**Minor points**:
- References need to be cleaned up -- several papers do not have the proper venue / journal listed.
- You don't seem to specifically define what is a small vs large dataset anywhere (terminology used in section 5.2, first paragraph, 4th point)

------------------------------------------------------------------------------------------------------------
[3] Edward J Hu, Yelong Shen, Phillip Wallis, Zeyuan Allen-Zhu, Yuanzhi Li, Shean Wang, and Weizhu Chen.  Lora:  Low-rank adaptation of large language models.

[4] Shuhuai Ren, Yihe Deng, Kun He, and Wanxiang Che. Generating natural language adversarial examples through probability weighted word saliency. In Proceedings of the 57th annual meeting of the association for computational linguistics, pp. 1085–1097, 2019.

**Strength And Weaknesses:**

**Strengths**
- The work is very thorough with many ablations to illustrate the benefits from the different modeling decisions suggested by the authors
- There are several novel contributions, in particular the Hotprotein benchmark and the structure-aware pre-training. The work also borrows ideas introduced in other fields to address (in a novel way) problems in protein prediction models and protein editing.

**Weaknesses**
- The main flaw I see in this paper is with respect to model performance validation. Unlike in many other application domains, protein sequences are not iid samples from the underlying data distribution as there is a lot of dependency between sequences that share common ancestors. This calls for particular cross-validation schemes to mitigate data leakage between training and test sets (eg., ensuring that sequences are not too similar), especially in several of the (semi-)supervised settings discussed in this work. See for example [1] for some ideas on how to do this. Similarly, it would be helpful if authors could comment on the similarities / overlap between the introduced Hotprotein dataset and the other datasets used in fine tuning / editing experiments (eg., FireProt and Meltome Atlas) -- if no substantial overlap, could you please specify that in supplementary?
- The paper is not very clear at times. For instance, the reasoning / rationale that gave rise to the equations 1 and 2 in section 4.1 is absent
- Not really a weakness, but rather a missed opportunity: since the SAP and fine-tuning frameworks are general and agnostic to thermostability, more extensive validation across protein families & fitness / protein attributes could be carried out to validate generalizability of the introduced ideas beyond thermostability. See for instance large-scale fitness benchmarks like ProteinGym [2].

------------------------------------------------------------------------------------------------------------

[1] Magńus Halld́or Ǵıslason,  Felix Teufel,  Jośe Juan Almagro Armenteros,  Ole Winther,  and Henrik  Nielsen.   Protein  dataset  partitioning  pipeline.   2021b.   URL https://github.com/graph-part/graph-part

[2] Notin, P., Dias, M., Frazer, J., Marchena-Hurtado, J., Gomez, A.N., Marks, D.S., & Gal, Y. (2022). Tranception: Protein Fitness Prediction with Autoregressive Transformers and Inference-time Retrieval. ICML.

**Summary Of The Paper:**

This paper focuses on the prediction of protein thermostability and the design of more thermostable proteins. After introducing a new benchmark to train and assess thermostability prediction models, authors describe a contrastive-learning framework to impart protein structure information on representations learned by large transformer networks usually trained on sequence alone (eg., ESM-1b). The paper also adapts ideas from other domains to carry out robust fine tuning and protein editing.

**Summary Of The Review:**

Overall a very thorough and interesting work. Some concerns on validation as discussed in the strengths / weaknesses section. I am leaning accept overall given the several meaningful contributions, in particular the HotProtein dataset and the structure-aware pre-training. I would be willing to recommend acceptance more enthusiastically if the aforementioned concerns are addressed during rebuttal.

--------------------------------------------------------------------------------------------------------
[UPDATES POST REBUTTAL]

Thank you to the authors for their thorough responses during rebuttal. My main concerns have been adequately addressed (in particular regarding validation / potential data leakage). I have also read in detail the assessments from the other two reviewers, and believe their comments have also been properly addressed in reviews. I do believe that the newly introduced HotProtein benchmark, together with the methodological contributions (structure aware pre-training) and various adaptations from other domains to protein modeling (eg., factorized fine tuning and feature augmentations) represent a solid contribution. In my opinion this paper should be accepted to the conference, and I have increased both my score for correctness and overall appreciation accordingly.

---

> ### Author Response · Authors · 2022-11-19
> **Response to Reviewer Uvn5 [Cons 1-3]**
>
> Many thanks to reviewer Uvn5 for acknowledging our experiments are “thorough”, our proposals are “meaningful/novel”, and our paper is “well written and easy to follow”. We sincerely appreciate all constructive suggestions, which help us to improve our paper further. To address reviewer Uvn5’s questions, we provide pointwise responses below.
>
> **[Cons 1. Model Performance Validation.]** Thanks for the helpful suggestions and references. We offer responses in the following three aspects.
>
> *<Sequence Similarity in Hotprotein.>* As indicated in Section 3, during the preprocess of HotProtein, we utilize CD-Hit (http://weizhong-lab.ucsd.edu/cd-hit/) to cluster protein sequences across organisms within a “TemperatureRange” class at a sequence similarity threshold of 50%. In each cluster, we only keep one protein whose sequence length is between 200~500. Therefore, HotProtein has already removed many “redundant” sample sequences, with a similar clustering-based approach to [1].
>
> *<Additional Evaluation on Hotprotein with New Train-Test Splits.>* Moreover, to further address reviewer Uvn5’s concern, we conduct additional evaluations with new train-test splits beyond the cross-validation. Specifically, we randomly sample 10% of proteins as a test set and treat the rest 90% of proteins as the training set. Note that any samples from the new test set have a sequence similarity below 50% to any sequence in the training set. Under this new evaluation scheme, our methods obtain 89.42% accuracy and 82.04% spearman correlation on HP-S, i.e., 3.17% accuracy boosts and 6.65% correlation gains compared to the ESM-1B baseline. Observations and conclusions remain the same.
>
> *<Sequence Similarity Between Hotprotein and FireProtDB.>* As specified in Section E of our appendix, the sequence similarities between HotProtein and FireProtDB are low. To be specific, we use the classic tool, i.e., CD-Hit, to compute the protein sequence similarity (1) between FireprotDB and Hotprotein; (2) within FireprotDB. We find the similarity between FireportDB and Hotprotein (0.1928) is much lower than the one within FireprotDB (0.2504). It suggests the substantial domain shift between Hotprotein and FireProtDB.
>
> Moreover, the task discrepancy between temperature prediction on HotProtein and editing suggestion on FrieProtDB is significant, even for the same protein sequences. The former only models the temperature of target proteins, and has no knowledge about the temperature change given single-point mutations. However, the latter focuses on modeling the temperature changes from single-point mutation.
>
>
> **[Cons 2. The Reasoning and Rationale of Equations 1 and 2 in Section 4.1.]** Due to the space limitation, we describe the motivations in Appendix B. We would add a hint to the appendix in the revision. In short, the motivation is to enhance the whole protein representation by contrastive learning, instead of directly using the amino acid representations. Once we target the global representation of one sentence (or one protein), we should learn from a loss function that directly compares one sentence and another (Gao et al., 2021). Similar to the sentence representations of BERT (Devlin et al., 2018), we notice that the protein representations learned by sequence models are not uniformly distributed in the latent space (Gao et al., 2021). Therefore, we adopt contrastive learning in the protein embedding space to distribute the representations more uniformly and let the 3D model inject information into the sequence model.
>
> **[Cons 3. Writing Issues and Other Minor Points.]** Thanks for such detailed feedback. We have addressed them in our updated draft.
>
> *<More Accurate Language.>* We have changed the terms to “adopt”, “introduce”, or “customize”.
>
> *<Clean up the Reference Format.>* References have been cleaned up with the properly listed venue and journal instead of the preprint versions.
>
> *<Definition of Small versus Large Dataset.>* Thanks. We have clarified this in both Sections 3 and 5 of our revision.

---

> > ### Comment · Reviewer_Uvn5 · 2022-12-12
> > **Re: author responses**
> >
> > Dear authors,
> >
> > Thank you for your thorough responses during rebuttal. My main concerns have been adequately addressed (in particular regarding validation / potential data leakage), and I have updated my score accordingly. I have added concluding thoughts at the end of my original review.

---

> ### Author Response · Authors · 2022-11-19
> **Response to Reviewer Uvn5 [Interesting Extension]**
>
> **[Interesting Extension. Validate Generalizability Beyond Thermostability.]** Yes, as acknowledged by reviewer Uvn5, ideally our algorithmic framework is generalizable to any protein phenotype, for which we have sufficient experimental data. We are looking into deep mutational scanning datasets where they characterize the fitness of numerous proteins. Thank you for pointing us to ProteinGym, we were in the middle of building our own data pipelines to do a DMS analysis with our pre-training and fine-tuning framework. This saved us a lot of time and enabled us to carry out new experiments and benchmark our model performance on ProteinGym. Our model’s improved performance on ProteinGym validates the generalizability of our methods beyond modeling thermostability.
>
> Specifically, we randomly sample 50,000 single-point mutation results from the Protein Gym as the training samples and another unseen 10,000 for the testing samples. Experiments are conducted and their testing performance is collected in below Table R1.
>
> Table R1. Performance comparisons of the sampled single point mutation result from Protein Gym. We examine two methods, ESM-1b and ours (ESM + FST + ADV + SAP).
> | Method | Spearman | Pearson |
> | :----: | :----: | :----: |
> | ESM-1b | 0.7203 | 0.7258 |
> | Ours | 0.7372 | 0.7311 |

---

> > ### Comment · Reviewer_Uvn5 · 2022-12-12
> > **Re: extension to fitness prediction**
> >
> > Thank you for sharing - very promising results to further illustrate the generalizability of the method. Would recommend that the train-test split to be done by protein family instead (ie., train on mutations from a handful of families, test on other assays) -- instead of randomly sampling mutations -- to avoid the issues with data leakage discussed during rebuttal (mutants in DMS assays are relatively close to one another). Would be very curious to see the performance lift breakdown to understand which components contribute most to the increase (FST Vs Aug Vs SAP)

---

### Decision · Program_Chairs · 2023-01-20

**Decision:**

Accept: poster

**Justification For Why Not Higher Score:**

The paper is on the borderline on the reviews, so the question is between accept / reject, but not higher.

**Justification For Why Not Lower Score:**

This is a borderline; unfortunately, due to the limitations of the time zones, organizing the in-person meeting was close to impossible.
I think, the authors have completely answered the concerns of the reviewers (even with the reject scores), while reviewers did not comment on the rebuttal. My personal reading (although I am not a biology expert at all)  gives the impression that this is a good paper. I personally liked the description of the dataset creation, which reads more like a novel.
The part with the novel and factorized learning is really interesting.
All of these different parts make the model quite solid, and taking into account the amount of concurrent work I think it would be a pity if the authors did not not make it into the conference this time - this work may just loose to the competitors just on the random reject.

**Metareview: Summary, Strengths And Weaknesses:**

The paper has two parts: the creation of the dataset ("Hot protein") focusing on protein thermostability
and the novel learning framework which uses structure-aware pretraining.

The strength of the paper is both the dataset and the training framework, which borrows ideas from the other fields into the protein domain, which suffers from the lack of large datasets (typical for other fields). The part with factorized weights is really interesting, since it shows a practical way to learn generalizable models with limited data available.


The main concern that is present in the reviews is the model evaluation. The data is not i.i.d by default, and cross-validation can be tricky. However, the authors have given a complete answer to this question by clustering proteins and keeping one sequence in the cluster. The "reject" by other reviewer had the same concern, so I consider that this question is answered (the reviewer did not engage in the discussion).
The other possible weakness could be the missing baselines, but the authors answered this as well in their rebuttal. ]

Overall, I think it is a good paper, which is well-written.




**Note From Pc:**

if the above contains the word "oral" or "spotlight" please see: "oral" presentation means -> notable-top-5% and "spotlight" means -> notable-top-25%. As stated in our emails, we are disassociating presentation type from AC recommendations

**Summary Of Ac-Reviewer Meeting:**

N/A